



# Impact of mixing state and hygroscopicity on CCN activity of biomass burning aerosol in Amazonia

Madeleine Sánchez Gácita[1], Karla M. Longo[1], Julliana L. M. Freire[1], Saulo R. Freitas[1], Scot T. Martin[2]

[1]Center for Weather Forecasting and Climate Research, INPE, Cachoeira Paulista, SP, Brazil
[2]School of Engineering and Applied Science, Harvard University, Cambridge, MA, USA

*Correspondence to*: M. Sánchez Gácita (madeleine.sanchez@cptec.inpe.br)

**Abstract.** Smoke aerosols prevail throughout Amazonia because of widespread biomass burning during the dry season. External mixing, low variability in the particle size distribution and low particle hygroscopicity are typical. There can be profound effects on cloud properties. This study uses an adiabatic cloud model to simulate the activation of smoke particles

as cloud condensation nuclei (CCN) and to assess the relative importance of variability in hygroscopicity, mixing state, and activation kinetics for the activated fraction and maximum supersaturation. The analysis shows that use of medium values of hygroscopicity representative of smoke aerosols for other biomass burning regions on Earth can lead to significant errors, compared to the use of low hygroscopicity reported for Amazonia. Kinetic limitations, which can be significant for medium and high hygroscopicity, did not play a strong role for CCN activation of particles representative of Amazonia smoke

aerosols, even when taking into account the large aerosol mass and number concentrations typical of the region. Internal compared to external mixing of particle components of variable hygroscopicity resulted in a significant overestimation of the activated fraction. These findings on uncertainties and sensitivities provide guidance on appropriate simplifications that can be used for modeling of smoke aerosols within general circulation models.

## 1 Introduction

Aerosol-cloud interactions are a major source of uncertainties in the quantification of climate forcing of aerosols (Bauer and Menon, 2012; IPCC, 2013). The wet size of an aerosol particle when at equilibrium with the environment is governed by Köhler theory (McFiggans et al., 2006) and depends on particle size and composition. In the atmosphere, activation as cloud condensation nuclei (CCN) is a competition between aerosol particles for water vapor, influenced by dynamical processes and the kinetics of particle growth and dependent on the updraft velocities, aerosol number concentrations and differences in

size and composition of aerosol particles (McFiggans et al., 2006). Although our understanding of the processes involved in aerosol activation has increased considerably in recent years (Farmer et al., 2015), the inclusion of all the detailed information that might be available about aerosol populations into global and regional circulation models is often impractical. Thus, assessments of the uncertainties derived from simplifications assumed are relevant and potentially





contribute to the discussion on the level of sophistication required by general circulation models (GCMs) with the aim of decreasing the uncertainties.

A large quantity of aerosol particles is generated globally by open biomass burning (Granier et al., 2011; Lamarque et al., 2010; van der Werf et al., 2010), and the impacts of smoke aerosols in climate, air quality and geochemistry have being addressed in several studies (Andreae, 1991; Crutzen and Andreae, 1990; Jacobson, 2004; Langmann et al., 2009; Tosca et al., 2013, and references there in). Vegetation fires plumes can be entrained into upper levels of the troposphere and undergo long-range transport before being removed from the atmosphere if conditions are favorable, e.g. when convection activity is high, (Andreae, 1991; Andreae et al., 2001; Freitas et al., 2005; Fromm and Servranckx, 2003). During the dry season in South America, observation and numerical model results agree in that biomass burning aerosol originated from extensive fires typically detected over the Amazon and Central Brazil regions, represents a significant fraction of the aerosol burden in South and Southeast parts of Brazil, Uruguay and the Northern of Argentina (Camponogara et al., 2014; Freitas et al., 2005; Longo et al., 2010; Ramanathan, 2001; Rosário et al., 2013; Wu et al., 2011).

Even though a large fraction of biomass burning aerosols has low to moderate hygroscopicity (Carrico et al., 2010; Dusek et al., 2011; Engelhart et al., 2012; Petters et al., 2009; Rissler et al., 2006), biomass burning particles can act as CCN under sufficiently high atmospheric water vapor supersaturations (Mircea et al., 2005; Rose et al., 2010; Vestin et al., 2007). Therefore, CCN activation properties of pyrogenic particles are likely to be relevant for the aerosol climate forcing.

Some external mixing in terms of hygroscopicity seems to be rather common in aerosol populations, particularly over continents (Kandler and Schütz, 2007; Swietlicki et al., 2008). Yet, average hygroscopicity parameters have been estimated assuming internal mixing for aerosols from the same emission source (e.g., biomass burning), or even within the same geographical region (Gunthe et al., 2009; Pringle et al., 2010), and often used in GCMs. Sensitivity of CCN activation to hygroscopic mixing state under equilibrium conditions is also significant, and the assumption of total internal mixing could result in an overestimation of the CCN population that can range from 10% to 100% (Cubison et al., 2008; Ervens et al., 2010; Padró et al., 2012; Wex et al., 2010). The impact of mixing state under dynamic conditions has, however, been less studied, and some evidence suggests that conclusions from equilibrium conditions might not be directly extrapolated to CCN activation during cloud formation (Cubison et al., 2008; Ervens et al., 2010).

Although the effects of composition on the cloud droplet number concentrations are typically secondary when compared to those of population number concentration and size distribution (Dusek et al., 2006; Feingold, 2003; Hudson, 2007; McFiggans et al., 2006; Reutter et al., 2009), the extent to which its complexities can be safely neglected in GCMs is also yet to be established. Droplet number concentrations were shown to be more sensitive to the presence of organic content than to the updraft velocity in some situations (Rissman et al., 2004). On conditions typical of pyrocumulus (number concentrations up to $10^5$ cm$^{-3}$ and updraft velocities up to 20 m s$^{-1}$), Reutter et al. (2009) found that cloud droplet number concentration was sensitive to compositional effects (hygroscopicity). For three different ratios of the aerosol number concentrations to the updraft velocity, and for a fixed aerosol size distribution, the authors found that the sensitivity to hygroscopicity was low for medium to high hygroscopic values, but moderate for very low and low hygroscopicity values (Reutter et al., 2009). Still,





sensitivities to hygroscopicity are likely to be tightly related to the position of the dry critical size of the smallest activated particle within the overall size distribution of the aerosol population, and significant sensitivities have been obtained for the population of small aerosol particles with medium and high hygroscopicity (Ward et al., 2010).

Aerosol particles with critical supersaturations smaller than the maximum supersaturation reached within the cloud can nonetheless become interstitial aerosols due to the evaporation and deactivation mechanisms described by Nenes et al. (2001). These kinetic limitations, sometimes neglected in GCMs, are expected to be large when significant aerosol loads are present (Nenes et al., 2001). Consequently, parameterizations that assume equilibrium conditions overestimate CCN when kinetic limitations are important (Nenes et al., 2001; Phinney et al., 2003). However, little is known about how kinetic limitations are related with the particle hygroscopicity, although a relation between the timescale of the components solubility and activation has been reported (Chuang, 2006).

In the present study, we used an adiabatic cloud model to simulate the CCN activation of biomass burning particles, aiming to contribute to the understanding of the possible impact of different hygroscopicity values, mixing state and kinetic limitations in the CCN activated fraction. The modeling approach followed is described in Sect. 2. In Sect. 3, the observational findings for biomass burning aerosols in the Amazon region, as determined at ground sites during LBA-CLAIRE (Large-Scale Biosphere Atmosphere Experiment in Amazonia - Cooperative LBA Airborne Regional Experiment, 2001) (Rissler et al., 2004), LBA-SMOCC (Smoke Aerosols, Clouds, Rainfall, and Climate, 2002) (Rissler et al., 2006) and SAMBBA (South American Biomass Burning Analysis, 2012) (Brito et al., 2014) field campaigns in the Amazon region, are reviewed. According to the available observations, three typical situations in terms of size distributions and other aerosol parameters were considered in the definition of the case studies, described in Sect. 4. Finally, the results from the cloud parcel model and our conclusions are discussed in Sect. 5 and Sect. 6.

## 2 Modeling approach

### 2.1 Hygroscopicity

Several parameters have been proposed to describe the hygroscopic properties of aerosol particles at both sub- and supersaturated regimes (Rissler et al., 2010). One of such parameters, the effective hygroscopicity parameter $\kappa$ proposed by Petters and Kreidenweis (2007), hereafter called $\kappa_P$, was selected for this study. Using $\kappa_P$, the Köhler equation relating the particle wet size, $d$, and the water vapor saturation ratio at equilibrium with the particle, $S_{eq}$, takes the form (Petters and Kreidenweis, 2007):

$$S_{eq} = \frac{d^3 - d_{dry}^3}{d^3 - d_{dry}^3 \left(1 - \kappa_p\right)} \exp\left(\frac{A}{d}\right) \qquad (1)$$





where $A$ and $d_{dry}$ denote the Kelvin term and the particle dry diameter, respectively. For nomenclature of symbols used, the reader is referred to Appendix A.

The effective hygroscopicity parameter $\kappa_p$ has been extensively used after its proposition, and its value for several compounds and aerosol populations has been estimated (Almeida et al., 2014; Lathem et al., 2011; Petters and Kreidenweis, 2007). The relation between $\kappa_p$ and other parameters used to describe the aerosol water uptake properties can be found in Appendix B. According to the value of $\kappa_p$, the following categories have proposed by Gunthe et al. (2009): very low hygroscopicity (VLH, $\kappa_p < 0.1$), low hygroscopicity (LH, $0.1 \leq \kappa_p < 0.2$), medium hygroscopicity (MH, $0.2 \leq \kappa_p < 0.4$) and high hygroscopicity (HH, $\kappa_p \geq 0.4$).

## 2.2 Mixing state

The hygroscopicity parameter of an internal mixture of multiple components, assuming the Zdanovskii–Stokes–Robinson (ZSR) relation applies, is (Petters and Kreidenweis, 2007) can be estimates as $\kappa_p = \sum \kappa_{p_h} \chi_h$, where $\kappa_{p_h}$ and $\chi_h$ are the hygroscopicity parameter and volume fraction of the hygroscopic group $h$, respectively. For the same particle size, the volume fractions can be replaced by number fractions. Effective hygroscopicity parameters can be estimated for size-ranges and for the whole population from the values obtained for each size and hygroscopic group (Gunthe et al., 2009):

$$\kappa_{p_{eff}} = \sum \kappa_{p,group} f_{group} \tag{2}$$

where $f_{group}$ represents the group number fraction in the total aerosol.

From now on, $\kappa_p$ will denote the hygroscopicity of a single particle while $\kappa_{p_{eff}}$ will denote the population effective hygroscopicity parameter, equal to the particles $\kappa_p$ in an internal mixture or estimated according to Eq. (2) for an external mixture.

## 2.3 Cloud parcel model

A model of an air parcel assumed to ascend adiabatically at a prescribed updraft velocity and without entrainment to supersaturation conditions was used to study the activation of aerosol particles in the first stages of cloud development. The air parcel model used in this work is based on the model described by Pruppacher and Klett (Pruppacher and Klett, 1997), with the supersaturation and liquid water mixing ratio tendencies estimated as in Seinfeld and Pandis (2006) and the equilibrium supersaturation calculated as proposed by Petter and Kreidenweiss (2007). The aerosol dry size distribution for each hygroscopic group is discretized into $n$ bins with a fixed volume ratio for all bins. Particles that belong to bin size $i$ and hygroscopic group $h$ are assumed to grow equally when exposed to the same conditions. Coagulation and coalescence





processes are not considered, so the number of particles in each bin remains constant while their wet sizes change over time (full-moving size structure) (Jacobson, 2005).

The rate of change of the cloud droplet size, assumed to be only due to diffusional growth or evaporation, is determined by the expression:

$$\frac{d\, d_{i,h}}{d\,t} = \frac{4G}{d_{i,h}} \left( s - s_{eq} \right).$$ (3)

where $s$ is the air parcel supersaturation, $d_{i,h}$ and $s_{eq}$ are the wet diameter and equilibrium supersaturation of particles in the bin $i$ and hygroscopic group $h$, and the size dependent growth coefficient $G$ is defined in Appendix C. The equilibrium supersaturation is calculated from the saturation ratio expressed in Eq. (1):

$$s_{eq} = \frac{d_{i,h}^3 - d_{dryi,h}^3}{d_{i,h}^3 - d_{dryi,h}^3 \left( 1 - \kappa_{p,h} \right)} \exp \left( \frac{4\sigma_{w/a} M_w}{R\, T \rho_w\, d_{i,h}} \right) - 1$$ (4)

where, similarly, $d_{dryi,h}$ is the dry diameter of particles in bin $i$ and group $h$, and $\kappa_{p,h}$ is the specific hygroscopicity parameter of particles in the group $h$. Equation (4) is also used to calculate the wet diameters at the atmospheric conditions at the initial relative humidity in the beginning of the simulation, when particles are assumed to be in equilibrium with their environment.

The supersaturation rate of change is given by

$$\frac{d\,s}{d\,t} = \alpha(T)\,W - \gamma(p,T)\,\frac{dw_L}{d\,t}$$ (5)

where $W$ is the cloud parcel updraft velocity, and definitions for size-independent coefficients $\alpha$ and $\gamma$ can be found in Appendix C.

The rate of change of the liquid water mixing ratio $w_L$ for a population of droplets was estimated using the following expression:

$$\frac{dw_L}{d\,t} = \frac{\pi}{2} \frac{\rho_w}{\rho_a} \sum_{h=1}^{hgroups} \sum_{i=1}^{n} N_{i,h} d_{i,h}^2 \frac{d\, d_{i,h}}{d\,t} \quad .$$ (6)

The pressure is estimated assuming the environment is in hydrostatic equilibrium, and the temperature and water vapor mixing ratio are estimated from the moisture and heat conservation, respectively (Pruppacher and Klett, 1997). The surface tension dependence on temperature is relevant to CCN activation (Christensen and Petters, 2012), and it is calculated as

$\sigma_{w/a} = 7.61 \times 10^{-2} - 1.55 \times 10^{-4} \left( T - 273.15 \right)$ (Seinfeld and Pandis, 2006).

The cloud parcel model described was fully implemented in *Mathematica*® 10.0 (Wolfram Research, 2014). Equations (3), (5) and (6), together with the expressions for the air pressure, temperature, and water vapor mixing ratio, form a closed system of $n+5$ non-linear ordinary differential equations (ODE) in which derivatives depend not only on the set of variables





but on their derivatives as well. The ODE system was solved using IDA method from SUNDIAL package (SUite of Nonlinear and DIfferential/ALgebraic equation Solvers) (Hindmarsh and Taylor, 1999; Hindmarsh, 2000), as implemented in the function NDSOLVE of *Mathematica*.

### 2.4 Particle activation and kinetic limitations

Aerosol particles with wet size larger than their critical size are considered strictly activated as CCN. In this work, the particle's critical diameter is determined for each bin size and hygroscopic group as the value that maximized the particle's equilibrium supersaturation, given by Eq. (4). The total cloud droplet number concentration, $CCN_{neq}$, is estimated as the sum of strictly activated particles and those with wet sizes larger than activated particles. Particles larger than activated particles are considered cloud droplets as well because they have wet sizes larger than that of cloud droplets and can

condensate significant quantities of water vapor on their surfaces (Nenes et al., 2001). Unless otherwise stated, hereafter CCN will refer to $CCN_{neq}$, as estimated at the end of the simulation.

Many parameterizations of CCN used in GCMs assume that particles are in equilibrium with the environment until the maximum supersaturation is reached and consider as activated all particles with critical supersaturation less or equal to the air parcel maximum supersaturation. If the particle equilibrium supersaturation is expressed in its simplified form and

particles are assumed to respond instantly to changes in the air parcel supersaturation, particles with critical supersaturation lower than a given supersaturation $s$ will also have dry sizes larger than a dry particle cut diameter $d_{dry,c}$ (details in Appendix D). The cloud droplet concentration estimated thus, $CCN_{eq,max}$, effectively represent the maximum cloud droplet concentration attainable during the simulation. If evaporation and deactivation mechanisms of kinetic limitations (Nenes et al., 2001) are significant, the calculation of the CCN spectra from the maximum supersaturation assuming equilibrium will

lead to an overestimation of the CCN concentration number. In an intermediate approach, particles can be considered activated as CCN if their wet diameters are larger than the approximate cut wet diameter $d_c$ that corresponds to $d_{dry,c}$ in equilibrium conditions (Appendix D). This approximate estimation, denoted $CCN_{neq\_simp}$, considers kinetic effects to some extent since the wet sizes of particles that are compared to $d_c$ are calculated explicitly in the cloud model.

In order to measure the impact of kinetic limitations in the simulations, estimations by the three aforementioned methods are

presented. In addition, the ratio of $CCN_{neq}$ to the cloud droplet concentration obtained at equilibrium conditions, $CCN_{neq}/CCN_{eq,max}$, is estimated at the time of maximum supersaturation and at the end of the simulation.

### 2.5 Regimes of cloud droplet formation

This work follows the three-regime classification of Reutter et al. (2009) of CCN activation in a parcel ascending at a constant updraft speed. The first regime is an updraft-limited regime, in which the CCN activation is almost independent on





CN and the maximum supersaturation and CCN/CN are usually within the ranges $s_{max} < 0.2\%$ and CCN/CN $< 20\%$, respectively. The second is an aerosol-limited regime, in which the CCN is proportional to CN number concentration and only weakly dependent on $W$, with $s_{max} > 0.5\%$ and CCN/CN $< 90\%$. Finally, the third is a transition regime between the first two that is aerosol- and updraft-sensitive. Precise boundaries between these regimes were defined as those conditions

where the ratio between the relative sensitivities of the CCN to $W$ and to CN is equal to 4 or 1/4, respectively. For most conditions, $W/CN \approx 10^{-4}$ m s$^{-1}$ cm$^3$ and $W/CN \approx 10^{-3}$ m s$^{-1}$ cm$^3$, estimated for $\kappa_p = 0.2$, have been used as approximations to the borderlines between the regimes (Reutter et al., 2009; Ward et al., 2010).

## 2.6 Sensitivity of CCN to a parameter

Sensitivities $S(X_i)$ in the context of CCN activation were first introduced by Feingold (2003) as the slope in the linear

regression to the logarithms of cloud-top effective droplet radius $r_{eff}$ as a function of the logarithms of the parameter $X_i$, i.e. $S_{X_i} = \partial \ln r_{eff} / \partial \ln X_i$. Later on, McFiggans et al. (2006) proposed sensitivities of the drop number concentration (CCN) to a parameter $X_i$:

$$S_{X_i} = \frac{\partial \ln(\text{CCN})}{\partial \ln(X_i)} \tag{7}$$

According to Eq. (7), $\text{CCN} \propto X_i^{S_{X_i}}$, and a sensitivity closer to zero indicate a smaller increase in CCN as parameter $X_i$

increases. Sensitivities were calculated from linear regressions in $\ln(\text{CCN})$ vs. $\ln(X_i)$ curves as averages (slope of the linear fit) and locally (derivatives of the curves in the $\ln - \ln$ space).

## 3 Overview of biomass burning aerosols observations in Amazonia

Observations on the biomass burning aerosol size distribution, hygroscopic properties and mixing state in the Brazilian Amazonia available in the literature are reviewed in this section, aiming to substantiate afterward the definition of

hypothetical case studies that nonetheless reflects the characteristics of the smoke aerosol population in this region. This overview focus largely on four datasets of ground site observations, with analyzed periods ranging from some days to almost a month, which were conducted in the framework of three experiments: the Large-Scale Biosphere Atmosphere Experiment in Amazonia / Cooperative LBA Airborne Regional Experiment in 2001 (LBA/CLAIRE) (Rissler et al., 2004), the LBA / Smoke Aerosols, Clouds, Rainfall and Climate in 2002 (LBA/SMOCC) (Andreae et al., 2004) and the South American

Biomass Burning Analysis in 2012 (SAMBBA) (Brito et al., 2014).

The Amazonia climate and meteorological conditions during each of these experiments are briefly described in Sect. 3.1, while physical properties of the smoke aerosol are addressed in Sect. 3.2.



### 3.1 Regional conditions during the observations

The main large-scale systems affecting central Brazil and the Amazon Basin during the winter in the Southern Hemisphere are the Intertropical Convergence Zone (ITCZ), mid-latitude frontal systems, and the South Atlantic Subtropical High (SASH). The transition from the wet to the dry season comes with a tendency to a westward displacement of the SASH and

northward motion of the ITCZ. The dry season is then established during the austral winter with the SASH well settled over the continental South America and the ITCZ belt north the Equator, producing a high-subsidence area over the Amazon Basin, and displacing wetness and cloudiness to remote areas in the north and northwest Amazon. In addition, approaching cold frontal systems are usually blocked by the high-pressure system and driven eastward to the Atlantic Ocean. The dry season in Amazonia is a time with low values of accumulated precipitation and light easterly winds, favoring the occurrence

of vegetation fires. The transition from the dry to the wet season occurs with the weakening of this blockage and periodic penetrations of frontal systems northward, disturbing the atmospheric stability. Inter-annual phenomena, like El Niño Southern Oscillation (ENSO), also affect the climate pattern in Amazonia. As such, the following describes the specific characteristics of the observation periods.

### 3.1.1 LBA/CLAIRE experiment

During the LBA/CLAIRE experiment, observations of aerosol physical properties were acquired at a ground site surrounded by forested area in Balbina (1°55.2' S, 59°28.1' W), about 125 km northeast of Manaus.  Observations were conducted from July 4 to 28, 2001, during the transition from the wet to the dry season in Brazil. According to Rissler et al. (2004), during two periods of 4 and 3 days each, conditions at the ground site were, respectively, characteristic of 2.5 to 5 days old aged smoke (hereafter Aged BB period) and recent smoke (prevenient from dry grass burned at a community located 5 km up-

wind, hereafter, Recent BB period) biomass burning.

Average daily precipitation and precipitation anomaly for the data collection period of CLAIRE, as provided by the United States National Oceanic and Atmospheric Administration (NOAA) Climate Prediction Center CPC (Chen et al., 2008), are presented in Fig. 1 (a) and (b) panels, respectively. The mean daily precipitation typically ranged between 5 and 10 mm day$^{-1}$ in the northwestern and northern Amazonia. Meanwhile, the mean daily values were below 4 mm day$^{-1}$ in southern areas,

already decreasing toward dry season precipitation levels. The precipitation anomaly indicates that the period was, on average, wetter than the climatological mean for western Amazonia, though drier than some regions near the Brazilian northern border.

During the period covered by CLAIRE, no significant number of fires were detected nearby Balbina or upwind (INPE, 2015). The mean monthly mean value of aerosol optical depth (AOD) at 500 nm channel at the Balbina AERONET station

(Eck et al., 2003; Holben et al., 2001) was 0.08 (±0.03), while the precipitable water widely ranged between 3.5 -5.1 cm. Yet, during the Aged BB and Recent BB periods, values of AOD at Balbina were slightly higher, up to 0.13 and 0.14, respectively.





### 3.1.2 LBA/SMOCC experiment

Two datasets of observations of biomass burning aerosol were acquired during the LBA/SMOCC 2002 at Fazenda Nossa Senhora Aparecida (FNS, 10°45.73' S, 62°21,45' W) ground site, Rondônia, in the southwestern Amazon, during the dry season (11 Sep – 8 Oct) (hereafter DS period) and dry-to-wet transition period (9 Oct – 30 Oct) (DTW period) of 2002. The

area surrounding FNS ground site had experienced deforestation for more than two decades by the time of SMOCC2002, and is considered to be representative of southwestern Amazon, with a strong influence of biomass burning during the dry season (Andreae et al., 2002).

During the dry period of the SMOCC experiment, the mean daily precipitation was typically below 4 mm day$^{-1}$, lower than the climatological mean, for most regions in the North of Brazil. Yet some isolated areas showed precipitation of up to 10

mm day$^{-1}$, above the climatological mean for the period, mainly due to a cold front intrusion between 19 and 26 of September causing precipitation in the southern part of the Amazon region (Fig. 1, c and d). Meanwhile, during the dry to wet period, average daily rates were above 5 mm day$^{-1}$ in most of the northern region of Brazil. For this latter period, conditions were, on average, wetter than the climatological mean except for some areas in the south and southwest of Amazonia, which was an indication of the near start of the wet season (Fig. 1, e and f).

Until September 18, the dry and hot atmospheric conditions favored the occurrence of a high number of fires in the Brazilian Amazonian region, with September 18 as the day with the highest number of detected fires since August 1999 (CPTEC/INPE, 2002a). During this month, 61012 fires were detected by satellite NOAA 12 in Brazil in Brazil, many of which were concentrated in the south and southwestern of the legal Amazonia. Conditions for the first days of October, still in the DS period, were again dry and with high temperatures, and during this period up to 3000 fires were detected in Brazil

within a single day by the same satellite (CPTEC/INPE, 2002a, 2002b). The total number of detected fires in October was 49527, yet the highest numbers of fires per area were detected in the northwestern Amazonia and in the Northeast region of Brazil (INPE, 2015). At Abracos_Hill AERONET station, nearby the ground site, the monthly mean AOD values at 500 nm were 0.95 and 0.52, respectively, for September and October 2002.

### 3.1.3 SAMBBA experiment

More recently, in the context of the SAMBBA experiment, a set of ground observations were conducted in a ground site (8°41,4' S, 63°52,2' W) located on the border of forest inside a reservation, about 5 km north from Porto Velho (upwind the predominant wind direction), Rondônia. SAMBBA took place during the late dry season and the transition from the dry to the wet season in 2012. The dataset reviewed here refers to the period from 13 to 30 of September, in the transition from the dry to the wet season.

The mean daily accumulated precipitation during the period of observations was somewhat similar to that of the SMOCC dry period (Fig. 1, g and h), with an intense cold front incursion advancing up to the south and southwest of Amazonia. During this period of the SAMBBA experiment, the areas with positive precipitation anomalies were in larger in the western and





central Amazonia and in the east and northeast of Amazônia conditions were on average drier than in the dry period of SMOCC.

During September 2012, a total of 62,099 fire spots were detected with Aqua MT satellite but, unlike during the dry period of SMOCC in 2002, the higher number of spot fires were concentrated in the eastern and northeastern Amazonia (INPE, 5 2015). September 2012 average AOD at 500 nm in Porto_Velho_UNIR AERONET station was 0.49, comparable to that observed in 2002 for the transition period.

## 3.2 Biomass burning aerosols: size, hygroscopic properties and mixing state

Several observational biomass burning studies conducted in the Amazon region reported rather similar number size distributions for biomass burning aerosols within the boundary layer (Andreae et al., 2004; Artaxo et al., 2013; Brito et al., 10 2014; Reid et al., 1998; Rissler et al., 2004, 2006). For each of the three previously described experiments, 3 log-normal number size distributions were proposed to fit the average aerosol number size distributions observed during each period (Table 1). The geometric mean diameters in number size distributions for both Recent and Aged BB for CLAIRE, for Aitken (~ 70 nm) and accumulation mode (140 -150 nm), were similar to those adjusted for the data corresponding to the transition period in the SMOCC experiment (66 nm and 131 nm, respectively) and slightly smaller than those corresponding to the 15 average particle number size distribution for the dry period data of the SMOCC (92 nm/190 nm) and to the average data for the whole period of SAMBBA (~98 nm/~179 nm).

In CLAIRE and SMOCC studies, the hygroscopic behavior and CCN ability of smoke aerosols were also analyzed. In these two studies, the authors characterized the hygroscopic behavior using the parameters $\varepsilon$ and $\kappa_R$, and considered as reference salts ammonium hydrogen sulfate (*AHS*) and ammonium sulfate (*AS*), respectively (Rissler et al., 2004; Vestin et al., 2007). 20 In all periods from both CLAIRE and SMOCC, smoke particles were found to be externally mixed in terms of hygroscopicity (Rissler et al., 2004, 2006), but neither set of observations included smoke particles with medium or high hygroscopicity.

The parameters $\varepsilon$ and $\kappa_R$ for the biomass burning episode averages for CLAIRE and the afternoon averages for SMOCC, respectively, were converted to $\kappa_p$ as described in Appendix B, considering $\kappa_{pAHS} \approx 0.65$ and $\kappa_{pAS} \approx 0.62$ (Petters and 25 Kreidenweis, 2007). Diurnal values of the effective hygroscopicity parameter were also calculated for the dry season and the dry-to-wet transition period of SMOCC from the diurnal averaged H-TDMA growth factor data reported in Table 3 of Rissler et al. (2006). Population effective $\kappa_{p_{eff}}$ values estimated assuming internal mixing as described in Sect. 2.2 for hygroscopic groups and dry size ranges, are presented in Table 2. For SAMBBA, no H-TDMA data is available up to this date.

30 The differences between the aged biomass burning and the recent biomass burning episodes were very small for the aged BB and recent biomass burning periods in the CLAIRE study (~0.005 in absolute value of the population $\kappa_{p_{eff}}$) (Table 2), in





spite of the difference in terms of smoke age and origins, and probably also different fuel types and fire conditions. For the two periods of the SMOCC study, the values obtained for $\kappa_{p_{eff}}$ were in general low due to the predominance of a group with very low hygroscopicity. Afternoon averages of the hygroscopicity parameter were higher than diurnal averages for all size ranges and hygroscopic groups (up to a 0.04 absolute difference), and $\kappa_{p_{eff}}$ values during the dry-to-wet transition period were only slightly higher than values for the dry season (up to ~ 0.03 absolute difference). In addition, there was a slight tendency of larger particles to be more hygroscopic in all discussed observations, but differences in $\kappa_{p_{eff}}$ between the Aitken and accumulation modes were limited to ~ 0.02 for SMOCC while being more pronounced (0.03 to 0.06) for CLAIRE.

The effective hygroscopicity of particles in each size range (including particles from hygroscopic groups with very low hygroscopicity and low hygroscopicity) was largely driven by the relative abundance of each hygroscopic group. The fraction of aerosols with low hygroscopicity was predominant during CLAIRE (on average, 80%) and was surprisingly similar for both recent and aged biomass burning periods. Conversely, in the SMOCC study aerosols with very low hygroscopicity predominated for both dry and dry to wet transition periods. Aerosols with very low hygroscopicity were found more abundantly in the dry period than in the dry to wet transition period of SMOCC. On average, the very low hygroscopicity aerosols accounted for ~ 85% of the total aerosol in the dry period, and for daily and afternoon averages of 61% and 73%, respectively, in the dry to wet transition period. Further observations are still necessary to assess whether the VLH group is always more abundant in a more polluted environment, but these findings together suggest a relation between aerosol number concentration and the biomass burning aerosols aging process, i.e. a higher load of very low hygroscopicity particles in more polluted environments.

The very low and low values found for the hygroscopic growth factor and hygroscopicity parameter of smoke particles in Amazonia could be partly explained by their composition. Biomass burning aerosols in Amazonia are largely formed by organic carbonaceous material and, to a lesser extent, black carbon, with only smaller fractions of other inorganic trace species that could enhance the particles water uptake (Andreae and Merlet, 2001; Decesari et al., 2006; Fuzzi et al., 2007; Reid et al., 2005). While a $\kappa_p = 0.04 \pm 0.02$ has been previously suggested for freshly emitted (~minutes) biomass burning aerosol (Carrico et al., 2010), an average value of $\kappa_p = 0.10 \pm 0.02$ have been suggested for biomass burning secondary organic aerosol (SOA) based on chamber experiments, after hours of photochemical aging of smoke aerosols (Engelhart et al., 2012). An inverse relation between hygroscopicity and the ratio of mass concentrations of total carbon number (organic + inorganic) to mass concentration of inorganic ions the parameter has also been observed in controlled biomass burning experiments, i.e. a higher carbon content and/or a low concentration of inorganic can be associated to a lower hygroscopicity (Carrico et al., 2010). Likewise, a large fraction of the organic mass in biomass burning aerosols can be attributed to water-soluble organic compounds (Mayol-Bracero et al., 2002) and smoke particles might contain significant quantities of water soluble organic nitrogen (Mace et al., 2003), some of them surface active. Water-soluble organic compounds have, however,





limited solubility and can affect the hygroscopic behavior and CCN activity because their solubility and surface active properties (McFiggans et al., 2006; Mircea et al., 2005).

The $\kappa_{p_{eff}}$ values of Amazonian smoke aerosol compare well with observed values for biomass burning aerosols, but they are rather on the lower side of the range of values reported elsewhere. An average $\kappa_p = 0.21$ was obtained for a four days

biomass burning episode near Guangzhou, China (Rose et al., 2010). Reported $\kappa_P$ for freshly emitted smoke particles in biomass burning laboratory experiments reached values up to 0.6, although a significant amount of data indicated values between 0.02 and 0.2, with wood species and smoldering fires producing the less hygroscopic particles (Carrico et al., 2010; Dusek et al., 2011; Engelhart et al., 2012; Petters et al., 2009). A recent study of the hygroscopicity of smoke particles in Thailand reported ranging between 0.05-0.1 for $\kappa_P$ (Hsiao et al., 2016), similar to the values described in the studies

considered in this review.

## 4 Definition of case studies and simulation parameters

In this work, three hypothetical different size distributions were defined as a base for the cloud model simulations (Table 3). The case studies were defined aiming to explore the role of hygroscopicity and mixing state outside equilibrium conditions for biomass burning aerosols in Amazonia. Therefore, the parameters of the lognormal number size distribution were chosen

for the three cases as to resemble biomass burning aerosol observations in Amazonia (Table 1) while trying to minimize the impact of particle size.

CCN activation has been previously shown to be largely impacted by the geometric mean diameter of the aerosol number size distribution (McFiggans et al., 2006; Reutter et al., 2009; Ward et al., 2010), and the sensitivity of CCN to this parameter increases for smaller particle sizes (Ward et al., 2010). In the selection of the parameters for the lognormal size

distributions, the larger geometric mean diameters within the range of interest were thus favored. Also aiming to reduce the impact of particle size, as well as to ease the comparison between case studies, the geometric mean diameter and standard deviation were kept fixed for Aitken and accumulation modes, changing only the particle number concentrations in each mode. Particles in the nucleation mode were disregarded because, typically, they are not large enough to activate and they are not expected to impact significantly the CCN behavior of the aerosol population.

The number size distributions of the total population, Aitken and accumulation modes for each of the case studies are depicted in Fig. 2. The same particle number concentrations were chosen for high (5000 cm$^{-3}$) and low (1000 cm$^{-3}$) polluted conditions, aiming to improve comparability between the different cases. As discussed, the number of particles in Aitken and accumulation modes gives the differences between the three chosen cases. First, a moderated polluted case with 5000 cm$^{-3}$ particles in the Aitken mode, and 1000 cm$^{-3}$ in the accumulation modes, respectively (MP$_{5,1}$) (Fig. 2, a). Case MP$_{5,1}$ is similar

to the observed distribution during SAMBBA. Second, a case study with the same number concentration than MP$_{5,1}$, but with higher number of particles in the accumulation mode, with 1000 cm$^{-3}$ and 5000 cm$^{-3}$ in the accumulation and Aitken modes,





respectively (MP$_{1,5}$) (Fig. 2, b). The size distribution of case MP$_{1,5}$ is similar to the observed during SMOCC dry-to-wet transition period. There was also a predominance of particles in the accumulation mode during the biomass burning episodes of CLAIRE, although particle number concentrations were low for these periods. Finally, a highly polluted case (HP$_{5,5}$) (Fig. 2, c) with 5000 cm$^{-3}$ in both modes, resembling the observed distribution during the SMOCC dry period, minus the

nucleation mode.

To assess the role of aerosol mixing state outside equilibrium conditions, cloud model simulations were conducted for populations both externally and internally mixed. Results obtained for two hygroscopic groups of particles externally mixed are compared with results when assuming that the population is internally mixed. H-TDMA observations of biomass burning aerosols in Amazonia (Sect. 3.2) suggest that hygroscopic groups with very low and low hygroscopicity are ubiquitous for

smoke aerosols in this region, but they can be present at variable fractions. This situation was simulated as two hygroscopic groups having $\kappa_p = 0.04$ and $\kappa_p = 0.16$, respectively, with a population effective hygroscopicity given by Eq. (2), and was denoted *Ext1*. A second possibility, denoted *Ext2*, was considered to account for more hygroscopic biomass burning aerosols observed for other biomass/regions, and increased the $\kappa_p$ of the more hygroscopic group to a medium hygroscopicity value, $\kappa_p = 0.30$. The internally mixed population was denoted *Int*. The minimum/maximum $\kappa_p$ in both sets of externally mixed

populations is obtained for the extreme case when only one group is present (therefore reducing to the internally mixed case) and is equal to the hygroscopicity parameter of particles in this group.

The effective $\kappa_{p_{eff}}$ and the corresponding fractions of each group for both situations and different fractions of the hygroscopic groups are presented in Table 4. The schematic size distribution of the aerosol total population and that of the hygroscopic group with $\kappa_p = 0.04$ are indicated in Fig. 2 for the three study cases, for a $\kappa_{p_{eff}} = 0.10$ and *Ext2* external

mixing state. The aerosol composition was considered to be independent of particle size, assuming that the slight tendency of higher hygroscopicity of larger particles was typically not large enough to impact significantly the CCN behavior of the population. In order to analyze the effect of hygroscopicity to the CCN activation, the simulations conducted for the internally mixed population (*Int*) ranged from $\kappa_p = 0.02$ to $\kappa_p = 0.60$, for the defined MP$_{5,1}$, MP$_{1,5}$ and HP$_{5,5}$ cases. Simulations conducted for the externally mixed population (*Ext1* and *Ext2*) ranged between the minimum and maximum

$\kappa_{p_{eff}}$ (0.004 to 0.16 and 0.004 to 0.30, respectively).

Updraft velocities between 0.1 m s$^{-1}$ and 10 m s$^{-1}$ were considered. Higher number concentrations than considered here can be found in pyrocumulus, but it is probably safe to assume that their impact on the hydrological cycle and aerosol indirect effect on a regional scale is secondary when compared with that of the regional haze, so these extreme cases of polluted conditions were not covered in our study. According to the regimes proposed by Reutter et al. (2009) (Sect. 2.5), our study

focused largely on the aerosol-limited and aerosol- and updraft-sensitive regimes, with particle number concentrations that characterize polluted conditions like those found in the regional haze. For MP$_{5,1}$ and MP1,5 cases, the updraft limited case is





given approximately by $W \leq 1$ m s$^{-1}$, but the aerosol-limited is given by $W \geq 6$ m s$^{-1}$. For the HP$_{5,5}$ case, the approximate limit of the updraft limited case is given by $W \leq 1$ m s$^{-1}$, and the aerosol-limited by $W \geq 10$ m s$^{-1}$ (not considered in our simulations).

Cloud base initial conditions for the simulations were: temperature of 293 K, atmospheric pressure of 900 hPa and relative

humidity of 98%. To avoid unrealistic physical parameters, the final time of simulation was defined somewhat arbitrarily as the time required for the parcel to ascend 500 m at the considered updraft velocity. The parameters for the simulations are summarized in Table 5. The distribution was discretized into 1000 bins ranged from 15 nm to $10^4$ nm, leading to a relative error of less than 0.003% with respect to the log-normal distribution for all the cases considered in this study. To exclude particles that are not large enough to activate, only particles larger than 30 nm ($N_{30}$) were considered as cloud nuclei (CN)

in the calculation of CCN/CN fractions. For all the cases considered, the cloud nuclei larger than 30 nm fraction included almost all particles, with the lowest fraction $N_{total}/N_{30} = 0.994$ obtained for case MP$_{5,1}$.

## 5 Results and discussion

Maximum values of supersaturation and CCN activated fraction, as function of hygroscopicity, updraft velocity and mixing state, are presented in Fig. 3 for the various proposed case studies and mixing states. Due to the high CN number

concentrations that characterize polluted conditions in the three case studies, maximum supersaturations reached in the simulations were typically low and, except for the highest updraft velocities and for very low hygroscopicity values (VLH, $\kappa_p < 0.1$), with values that were below 0.5% in the MP$_{5,1}$ case, and below 0.4% in the MP$_{1,5}$ and HP$_{5,5}$ cases. The highest values of maximum supersaturation were obtained for the MP$_{5,1}$ case, with a majority of particles in the Aitken mode. Maximum supersaturations in this case were, in average, ~ 0.10% larger (absolute differences) than those obtained for MP$_{1,5}$

case, and about 0.15% higher than those obtained for HP$_{5,5}$ case. Meanwhile, the values of maximum supersaturation reached in the MP$_{1,5}$ case study were higher than those obtained in the HP$_{5,5}$ case, but slightly, with absolute differences between maximum supersaturation values of up to 0.05%, all else being equal, in spite of the much higher CN number concentrations in the latter case. The case study with the highest CN number concentration (HP$_{5,5}$) presented the largest CCN number concentrations. However, the largest CCN/CN fractions were instead reached in the MP$_{1,5}$ case, all else being equal. The

CCN/CN fractions for the HP$_{5,5}$ case were the lowest between all three cases for all values of $\kappa_p$ within the low hygroscopicity (LH, $0.1 \leq \kappa_p < 0.2$) and medium hygroscopicity (MH, $0.2 \leq \kappa_p < 0.4$) ranges, while for $\kappa_p$ in the VLH range the lowest CCN/CN fractions were obtained for the MP$_{5,1}$ case.

These results for the maximum supersaturations and CCN/CN fractions are explained by the Köhler theory, which predicts that the Kelvin term typically dominates the growing process for larger particles, while the Raoult term is more relevant for

smaller ones. Therefore, particles in the accumulation mode are likely to condensate water vapor on their surfaces more



readily than the comparatively smaller particles in the Aitken mode, growing larger and impacting more the maximum supersaturation reached than the latter. Moreover, the Raoult term is more significant the smaller the particle, thus the activation of particles in the Aitken mode is expected to be more altered by hygroscopicity than the activation of particles in the accumulation mode.

Among the variable parameters within the simulations, both maximum supersaturations and CCN/CN fractions were impacted the most by updraft velocity, for all study cases and mixing states. Mean sensitivities of CCN to $W$ in the $MP_{5,1}$, $MP_{1,5}$ and $HP_{5,5}$ study cases were, respectively, 0.66, 0.65 and 0.73, with very little variability with mixing state, as illustrated in Fig. 4 for $\kappa_{p_{eff}} = 0.10$. These mean values of $S_W$ are higher than previous estimations of 0.18 and 0.47 for clean ($< 1000$ cm$^{-3}$) and polluted (1000 cm$^{-3}$ to 3000 cm$^{-3}$) conditions, respectively, by McFiggans et al. (2006). Yet an

increase of the sensitivity to $W$ with the number concentration is consistent with the behavior expected within the updraft- and aerosol-sensitive regime that is, on average, the predominating regime. The adjusted $R^2$ coefficients in the linear fits of the $\ln(CCN)$ vs. $\ln(W)$ curves were $\geq 0.90$ for all cases and mixing states. However, the data points departed from the mean slope towards low and high updraft velocities for all case studies and mixing states (Fig. 4, top). CCN number concentrations were more sensitive (local $S_W$ up to 0.9) to increases in the updraft velocity for velocities within the updraft-

limited regime, while for the aerosol-limited regime the sensitivity to $W$ decreased to values between 0.1 and 0.4 (Fig. 4, bottom). This varying sensitivity to $W$ of the CCN number concentrations is in agreement with the changing behavior of CCN activation within each regime of CCN activation described by Reutter et al. (2009), that varies from a high sensitivity of activation with $W$ in the updraft-limited regime to almost no influence in the aerosol-limited one. The sensitivity of CCN to the aerosol number concentrations and the geometric mean diameter and standard deviation have been discussed

elsewhere (McFiggans et al., 2006; Reutter et al., 2009) and was not addressed here.

In contrast with $S_W$, the sensitivity to hygroscopicity $S_{\kappa_p}$ changed substantially with mixing state, and will be discussed in Sect. 5.3.

## 5.1 Aerosol mixing state

The aerosol mixing state modified both maximum supersaturations and CCN/CN fractions, although to different extents. The

values of maximum supersaturation were slightly underestimated for updraft velocities in the aerosol-limited and the aerosol- and updraft-sensitive regimes when internal mixing was assumed (Fig. 3, top). The absolute differences were up to ~0.01 % and ~0.03 % for the externally mixed *Ext1* and *Ext2* populations, respectively. For updraft velocities within the updraft-limited regime, however, the maximum supersaturation reached were lowest, and the values assuming an internal mixing were almost identical or marginally higher than those reached for externally mixed populations.





On the other hand, the internal mixing hypothesis typically led to overestimations in the CCN number concentrations, regardless of the somewhat lower values of maximum supersaturation reached for this mixing case. The effect of hygroscopic mixing state in the CCN behavior of aerosols can be illustrated through the consideration of an aerosol population with known size and composition but no information on the mixing state. According to the mixing rule, given by

Eq. (2), particles in the externally mixed population will have either larger or smaller hygroscopicity parameters than that of the internally mixed population average. The more hygroscopic groups in the external mixture will have smaller cut particle diameters and will activate more readily than the internally mixed particles. Consequently, the number of more hygroscopic particles that activates as CCN would be underestimated if internal mixing was presumed. Under the same assumption, the fraction of less hygroscopic particles that will be considered activated would be overestimated.

Although differences in activation for more and less hygroscopic particles due to internal mixing will contribute with opposite signs to the total CCN concentration number derived from mixing state, they are unlikely to cancel each other. The impact of mixing state in CCN number concentration is illustrated graphically in Fig. 5 for a specific case, were the schematic size distribution of particles that are activated as CCN in the $MP_{5,1}$, $MP_{1,5}$ and $HP_{5,5}$ case studies at a prescribed updraft velocity of $W = 5$ m s$^{-1}$ are presented for external and internal mixtures. For the selected simulation, in the externally

mixed population (*Ext2*) one hygroscopic group have $\kappa_p = 0.04$, in the VLH range, and is present in a fraction $f_{\kappa_p=0.04} =$ 0.77, while a second hygroscopic group have $\kappa_p = 0.30$, within the MH range, and $f_{\kappa_p=0.30} = 0.23$. Assuming internal mixing (*Int*), these two groups resulted in $\kappa_{p_{eff}} = 0.10$ (Table 4). The values of maximum supersaturations reached were somewhat lower when internal mixing state was assumed, between 2% and 3% depending on the study case. A fraction of particles in the MH hygroscopic group ( $\kappa_p = 0.30$ ) was indeed activated as CCN in the externally mixed *Ext2*, but was not considered as

CCN in the internal mixing, since the internally mixed population $\kappa_{p_{eff}}$ is lower and thus the cut size for activation in the internally mixed population is larger. However, an even larger fraction of the particles in the VLH group were not activated in the external mixing, but were considered as activated when internal mixing state was assumed. Thus, assuming internal mixing in this example, and characteristically in the conducted simulations, led to an overestimation of the CCN number concentration.

Box plots on top of data in Fig. 6 display the magnitude of the CCN overestimation for the range of updraft velocities, as well as the spreading of overestimations for different values of $\kappa_{p_{eff}}$, derived from the assumption of internal mixing state for the conducted simulations. The CCN overestimation was expressed as $CCN_{Int} / CCN_{Ext} - 1$, where $CCN_{Int}$ and $CCN_{Ext}$ refers to an assumption of internally and externally mixed population, respectively. CCN overestimations were larger when the module of the difference between the internal mixture $\kappa_{p_{eff}}$ and that of the hygroscopic group with closest value of

hygroscopicity in the external mixture was greater, i.e. when the internally mixed assumption was comparatively less valid. CCN overestimations close to the lower limit or below the interquartile range of CCN overestimations were obtained for





populations with fractions $f_{\kappa=0.16} \geq 0.67$ in the *Ext1* (with a resulting $\kappa_{p_{eff}} \geq 0.12$), and $f_{\kappa=0.30} \geq 0.62$ in the *Ext2* mixing ( $\kappa_{p_{eff}} \geq 0.2$). Within the aerosol- and updraft-sensitive regime, CCN overestimations were largest for all three cases. The larger number of particles in the Aitken mode in the $MP_{5,1}$ and $HP_{5,5}$ case studies resulted in larger overestimations in the CCN number concentrations even for the upper range of updraft velocities. In contrast, the CCN overestimations decreased

noticeably as the updraft velocity increased towards the aerosol-limited regime for the $MP_{1,5}$ case. Within the updraft-limited regime the typically low fractions of activated particles, as well as the estimations of $CCN_{Int} / CCN_{Ext} - 1$, were more susceptible to inaccuracies due to bin resolution.

Average overestimations for the externally mixed population *Ext1* were typically low, $5.7 \pm 2.4$ %, $5.1 \pm 2.1$ % and $2.9 \pm 2.0$ %, or the $MP_{5,1}$, $MP_{1,5}$ and $HP_{5,5}$ case studies. For population *Ext2*, and the same case studies, averages were slightly higher,

$12.4 \pm 4.7$ %, $10.4 \pm 4.5$ % and $10.5 \pm 3.8$ %, respectively. However, with particle number concentrations of 10 000 cm$^{-3}$ in $HP_{5,5}$ case, and 6000 cm$^{-3}$ in $MP_{5,1}$ and $MP_{1,5}$ case studies, the absolute overestimation ( $CCN_{Int} - CCN_{Ext}$ ) in the CCN number concentration was, respectively, $160 \pm 94$ cm$^{-3}$, $181 \pm 96$ cm$^{-3}$ and $224 \pm 137$ cm$^{-3}$ for *Ext1* simulations and $349 \pm 203$ cm$^{-3}$, $358 \pm 188$ cm$^{-3}$ and $467 \pm 272$ cm$^{-3}$ for the *Ext2*. Maximum absolute overestimations were reached for higher updrafts, for which the CCN/CN was higher for all mixing states. For *Ext1* simulations, the maximum absolute

overestimations were 304 cm$^{-3}$, 323 cm$^{-3}$ and 432 cm$^{-3}$ for the $MP_{5,1}$, $MP_{1,5}$ and $HP_{5,5}$ cases, respectively, while in *Ext2* simulations for the same study cases they were of 637 cm$^{-3}$, 642 cm$^{-3}$ and 838 cm$^{-3}$. These concentrations, although characterize polluted conditions like those that could be found in regional hazes in the Amazonia region, are still moderate in comparison with concentrations inside pyro-cumulus.

It is important to note that, would the maximum supersaturations achieved in simulations for both mixing states be the same,

the CCN number concentrations would be higher in the internal mixing case simulations and the CCN overestimations derived from assuming internal mixing would be larger. This difference in the achieved maximum supersaturations does not explains the much smaller impact of mixing state found for cloud parcel model results when compared to those obtained for equilibrium conditions and prescribed supersaturations, but is likely to contribute to it since, in the latter, the same maximum supersaturation is assumed in the estimation of CCN number concentrations for the different mixing states.

For Amazon smoke particles, these results indicate a CCN overestimation derived from assuming internal mixing overestimation that is below 10% for all conditions. However, biomass burning particles represent a significant fraction of the aerosol budget on a continental scale during the dry season and, considering the impact of mixing state with low hygroscopicity apparent in the results presented, to assume an internal mixture between these smoke particles and particles with medium or high hygroscopicity should be avoided.





## 5.2 Hygroscopicity

The behavior of the CCN activation, as hygroscopicity changed, was distinctly different for the different mixing states. When the population was assumed to be internally mixed, the mean average sensitivity to hygroscopicity, $S_{\kappa_p}$, was low for the case MP$_{5,1}$ (0.20), and very low for MP$_{1,5}$ (0.10) and HP$_{5,5}$ (0.12) case studies. These estimations are in good agreement

with those by Reutter et al. (2009) and Ward et al. (2010). For the externally mixed population, however, $\ln - \ln$ curves were far apart from a linear behavior and it was not possible to achieve linear fits. Obtained adjusted $R^2$ parameters were close to zero or negative and hence average sensitivities for externally mixed populations were not estimated.

Local sensitivities for the internal mixing state typically decreased as the hygroscopicity parameter increased, starting from median values of ~0.35 for the MP$_{5,1}$ case study and of ~0.20 for the MP$_{1,5}$ and HP$_{5,5}$ case studies (Fig. 7) until almost

stabilizing at values close to 0.15, 0.05 and 0.10 for the same cases for values of $\kappa_p$ within the medium and high hygroscopicity ranges. Notable exceptions were found within the updraft-limited regime for populations with high hygroscopicity where the impact of kinetic effects was high, as will be addressed later in Sect. 5.3. Except for cases within the updraft-limited regime, were kinetic limitations were significant, we found that for internally mixed populations and $\kappa_p$ within the MH or the HH ranges the impact of the hygroscopicity parameter in the CCN number concentrations was very

low, while for $\kappa_{p_{eff}}$ values within the VLH range the impact was low to moderate, in agreement with results obtained by previous studies (Dusek et al., 2006; McFiggans et al., 2006; Reutter et al., 2009; Ward et al., 2010).

On the other hand, the local $S_{\kappa_{p_{eff}}}$ for the externally mixed populations presented mean values (over results for different updraft velocities) that increased with $\kappa_{p_{eff}}$ from very low or even negative to values between 0.3 and 0.45 for the highest $\kappa_{p_{eff}}$ values (Fig. 7). This higher sensitivity of CCN number concentrations to $\kappa_{p_{eff}}$ in the external mixtures is also apparent

in the step increase of the CCN number concentrations obtained for the external mixing results for the larger average $\kappa_{p_{eff}}$ values (Fig. 3, bottom).

The increasing $S_{\kappa_{p_{eff}}}$ for external mixing cases can be illustrated through the consideration of the following example for the HP$_{5,5}$ case and an updraft velocity $W = 5$ m s$^{-1}$. In the internally mixed population with $\kappa_p = 0.30$, 62% of the total CN was activated as CCN. If the internally mixed population has, instead, $\kappa_p = 0.25$, the CCN/CN fraction is ~61%. However, if the

population with $\kappa_{p_{eff}} = 0.25$ is instead externally mixed, the fraction of particles with $\kappa_p = 0.30$ that reached activation increased to 67% but, of the particles with $\kappa_p = 0.04$ (19% of total population), only 22% reached activation. Consequently, even when the MH particles predominated, the resulting CCN/CN ratio was 58%, a more significant decrease from the case with $\kappa_p = 0.30$ than in the internally mixed population case.



Considering the results from the simulations and the little variability and low values of $S_{\kappa_{p_{eff}}}$ for internally mixed populations, variations of hygroscopicity within the MH and HR could be considered as rather secondary and neglected, especially if the difference in hygroscopicity is not large, since the level of sophistication within GCMs should be kept at minimum whenever the accuracy of results is not compromised. When the hygroscopicity is within the LH and VLH, however, the overestimation in the activated fraction might be substantial as illustrated in Fig. 8 for updraft velocities in the updraft- and aerosol sensitive regime, also for internally mixed populations. In the extreme case when $\kappa_p = 0.20$ was assumed for a population of $\kappa_p = 0.04$, the mean overestimation of the CCN population for the $MP_{5,1}$, $MP_{1,5}$ and $HP_{5,5}$ was, respectively, $54.3 \pm 3.7$ %, $22.4 \pm 1.4$ % and $26.6 \pm 2,3$ %. In comparison, if $\kappa_p = 0.60$ was presumed for aerosols with $\kappa_p = 0.20$, the mean overestimations in the CCN obtained for the $MP_{5,1}$, $MP_{1,5}$ and $HP_{5,5}$ cases and the same range of updraft velocities were, respectively, $15.5 \pm 1.6$ %, $4.8 \pm 0.3$ % and $6.4 \pm 0.8$%.

A significant overestimation of the CCN can thus result from assuming an hygroscopicity in the MH range for the Amazon smoke aerosols. These results suggest that larger values of $\kappa_p$ like those recomended for continental aerosol or biomass burning particles in other regions of the world are not adequate to describe the CCN behavior of Amazon smoke particles.

### 5.3 Kinetic limitations

Temporal series of the CCN activation with resolutions of 0.5 s and 1 s near the time of maximum supersaturation for strong and low to moderate updrafts, respectively, were used to analyze the particle growth and activation evolution in time. Three separate effects in the evolution of the CCN number concentration observed in the simulations for weak and sometimes even moderate updrafts that could be attributed to the effect of kinetic limitations: (1) a delay between the time when maximum supersaturation was reached and the time when the activated fraction is largest; (2) a decrease in the number of activated particles with cloud depth after the maximum activated fraction is reached; and finally, (3) a overestimation of the CCN if assuming equilibrium applies, $CCN_{eq}$.

The delay in activation was amplified with the increase of the particle $\kappa_{p_{eff}}$. A relation to particle size and number concentration was also apparent, being the delay longest for the $HP_{5,5}$ case, moderate in the $MP_{1,5}$ case, and much shorter for the $MP_{5,1}$ case, also for large $\kappa_{p_{eff}}$ values and weak updrafts. This is illustrated in Fig. 9 for an internally mixed population and $W = 0.5$ m s$^{-1}$. Due to the delay in activation, typically, a significant fraction of particles was not activated at the time maximum supersaturation was reached. Within the updraft-limited regime, the delay in the activation was such that at the time of maximum supersaturation no particles are activated for internally mixed populations with $\kappa_{p_{eff}}$ above a certain threshold. For an updraft velocity of $W = 0.5$ m s$^{-1}$, this threshold was $\kappa_{p_{eff}} = 0.50$ for the $MP_{5,1}$ case and $\kappa_{p_{eff}} = 0.35$ for the $MP_{1,5}$ and $HP_{5,5}$ case, respectively. In the $MP_{1,5}$ case, for an updraft velocity $W = 3$ m s$^{-1}$, already in the updraft- and aerosol



sensitive regime, the threshold was still $\kappa_{p_{eff}} = 0.35$. The maximum value of $CCN_{neq,simp}$ is also reached sometime after the maximum supersaturation is reached, and its value is slightly higher than the maximum of $CCN_{neq}$. However, strong kinetic effects obtained for the larger $\kappa_{p_{eff}}$ values near the time of maximum supersaturation for $CCN_{neq}$ are not so strong for $CCN_{neq,simp}$. After the maximum $CCN_{neq}$ is reached, however, differences between both estimations are below 1% and at

the end of the simulation both estimations are very similar. The fraction of particles not strictly activated in $CCN_{neq}$ is important only near the time of maximum supersaturation, indicating that this assumption has no influence in results presented in previous sections, were CCN concentrations were estimated at the end of the simulation. However, the differences near the time of maximum supersaturation would be larger if this fraction is disregarded.

For the externally mixed population *Ext1*, although $CCN_{neq}$ was significantly lower than $CCN_{neq}$ for weak updrafts, in all

the cases at least a fraction of particles was activated at the time of maximum supersaturation. For *Ext2* and $W = 0.5$ m s$^{-1}$, however, populations with $\kappa_{p_{eff}} \geq 0.12$, or $f_{\kappa_p = 0.30} \geq 0.31$, also showed $CCN_{neq} = 0$ for both MP$_{1,5}$ and HP$_{5,5}$ cases at the time of maximum supersaturation. This is exemplified in the Fig. 10 for three values of the effective hygroscopicity parameter. Interestingly enough, particles from both hygroscopic groups failed to activate in these conditions. The value of maximum supersaturation was very low in these cases and it is likely that particles in the more hygroscopic group

condensate the limited water vapor on their surfaces more readily, although not in enough quantities as to activate themselves, but limiting even more the water vapor available to less hygroscopic particles and preventing their activation as well. Particles from both groups seem to grow rather slowly and both groups appear to activate at the same time.

As moderate and strong updrafts were considered, the delay between maximum supersaturation and maximum activation reduced until no longer observed at the temporal resolution of the time series. Within the updraft limited regime, the mean

overestimation of $CCN_{eq,max}$ in comparison with $CCN_{neq}$ over the range of $\kappa_{p_{eff}}$, excluding those that led to $CCN_{neq} = 0$, ranged from ~10% to ~100% in internally mixed populations, and between ~10% to ~250% in externally mixed ones (Fig. 11). However, within the updraft- and aerosol-sensitive, the overestimation at the time of maximum supersaturation was below 12% in most situations while for $W \geq 6$ m s$^{-1}$, it was below 5% for all case studies and mixing states.

The overestimation in $CCN_{eq,max}$ at the time of maximum supersaturation can be explained by the evaporation mechanism.

As the cloud depth increases, and in particular at the defined end of the simulation, the deactivation mechanism can be more relevant. Although $CCN_{neq}$ was always lower at the end of the simulation that at its maximum, the difference was typically low, between 2% and 10% for most updraft velocities and mixing states, as evidenced in the similar the overestimations of both values by $CCN_{eq,max}$. Both evaporation and deactivation mechanisms were relevant for weak and even moderate updrafts, and a relation with particle size and number concentration was apparent, as previously reported by Nenes et al.

(2001) for ammonium sulfate particles (2001).





Our results show that the effects of kinetic limitations were strong when a significant fraction of particles with hygroscopicity in the MH or LH range was present. However, for particles with low and very low hygroscopicities like the Amazon smoke particles, kinetic limitations are unlikely to be important even if large aerosol loads are present. A relation between the time scale of solubility and the CCN behavior of aerosols have been known (Chuang, 2006). Yet, to the best of

our knowledge, this is the first time that kinetic limitations have been explicitly related to the population hygroscopicity.

## 6 Conclusions

The available data on smoke particles in the Amazon region (Sect. 3) suggest that an external mixing of two particle groups having very low and low hygroscopicity, respectively, is typical for this aerosol population. The effective hygroscopicity reported for the biomass burning aerosol population in this region, in particular when VLH particles predominated, is in the

lower range of $\kappa_p$ values reported for smoke aerosols worldwide (Carrico et al., 2010; Dusek et al., 2011; Engelhart et al., 2012; Hsiao et al., 2016; Petters et al., 2009). There appears to be weak or no dependence of hygroscopicity on particle size for Amazonia. Hygroscopicity between freshly emitted and aged aerosols is similar. There is variation, however, in hygroscopicity with aerosol mass concentration.

We conducted cloud model simulations using three hypothetical aerosol size number distributions that resembled the three

typical situations found in the literature for smoke aerosols in the Amazon, as well as different values of hygroscopicity and mixing state, and the impact of kinetic limitations in moderate to high polluted conditions. A low sensitivity of CCN number concentrations to $\kappa_{p_{eff}}$ was found for medium and large hygroscopicity when the population was internally mixed. Yet, the effective hygroscopicity of smoke particles for the Amazon appears to stand in the VLH and LH ranges, where the sensitivity to this parameter is moderate. For this range of $\kappa_{p_{eff}}$, the CCN population could be overestimated significantly if

larger values of hygroscopicity, like those suggested for biomass burning particles elsewhere, were to be used.

Hygroscopic mixing state in the conducted cloud model simulations led to differences lower than those obtained in previous studies that addressed mixing state for equilibrium conditions and prescribed supersaturations. In particular, the CCN overestimation was low for populations similar in hygroscopicity to the Amazon smoke aerosols (*Ext1* in the simulations), but higher when the external mixing was between groups with VLH and MH (*Ext2*). The $\kappa_{p_{eff}}$ parameter posed a much

larger impact on the CCN activation within the MH range for externally mixed populations than for internally mixed ones, even for low fractions of VLH aerosols. When $\kappa_{p_{eff}}$ is estimated assuming internal mixing, and in particular when particles of VLH are present, it is important to take into account that the typically low sensitivity to hygroscopicity of internally mixed populations does not apply and even relatively small variabilities in $\kappa_{p_{eff}}$ could affect the CCN behavior of the population. Consequently, assuming internal mixing of particles with very low and low hygroscopicity and particles with

moderate or large hygroscopicity should be avoided. Finally, kinetic limitations were found to be much lower for particles





within VLH and LH hygroscopic groups and, therefore, its impact on the CCN behavior of Amazon smoke particles is expected to be limited, even in the presence of large aerosol loads.

**Appendix A: Nomenclature of frequently used symbols**

$A$       Kelvin term

$a_w$       water activity

$B$       Raoult term

CCN       Cloud condensation nuclei

CN       Cloud nuclei

$CCN_{eq}$       CCN assuming equilibrium conditions

$CCN_{neq}$       CCN without assuming equilibrium conditions

$d$       cloud droplet diameter (particle diameter after water uptake)

$d_{dry}$       particle dry diameter

$d_{dry,c}$       particle cut diameter for activation (dry)

$f_{hg}$       number fraction of hygroscopic group $h$

$G$       size dependent particle growth coefficient

$G_f$       particle growth factor

$N_m$       mode number concentration

$R$       universal gas constant

$S$       saturation ratio

$S_{eq}$       equilibrium saturation ratio of a particle

$s$       supersaturation

$s_{eq}$       equilibrium supersaturation of a particle

$s_{max}$       cloud maximum supersaturation

$t$       time

$T$       temperature

$W$       cloud parcel updraft velocity

$w_L$       liquid water mixing ratio in the cloud parcel

$\alpha$       size-independent coefficient in the calculation of the supersaturation rate of change



| $\chi$ | volume fraction |
|---|---|
| $\varepsilon$ | aerosol soluble fraction |
| $\gamma$ | size-independent coefficient in the calculation of the supersaturation rate of change |
| $\kappa_R$ | hygroscopicity parameter by Rissler et al (2006) |
| $\kappa_P$ | specific hygroscopicity parameter by Petter & Kreidenweis (2007) |
| $\kappa_{P_{eff}}$ | population effective specific hygroscopicity parameter |
| $\nu$ | number of ions the solute dissociates into |
| $\rho$ | density |
| $\sigma_{w/a}$ | water/air interface surface tension |
| $\upsilon$ | dissociation factor of a solute |

**Subscripts**

$i = 1,...,n$    bins

$h = 1,...,hgroups$  hygroscopic group

**Appendix B: Relation between $\kappa_p$ and other parameters related to particle hygroscopicity**

The hygroscopic growth of aerosol particles at sub-saturated conditions is usually characterized by the diameter growth factor, defined as $G_f = d/d_{dry}$. $G_f$ can be determined using a Hygroscopic Tandem Differential Mobility Analyzer (H-TDMA), which additionally offers information of the hygroscopic mixing state of the aerosol population.

Rearranging Eq. (1) it can be showed that $\kappa_p$ is related to $G_f$ by the relation

$$\kappa_p = \left(G_f^3 - 1\right)\left(a_w^{-1} - 1\right) \tag{B1}$$

The following parameters also describe the water uptake of aerosol particles: the soluble volume fraction $\varepsilon$ and the also named $\kappa$ parameter (Rissler et al., 2004; Vestin et al., 2007), from now on called $\kappa_R$. Representing the aerosol particle as an insoluble core and a soluble fraction assumed to be consisting of a reference salt, Rissler et al. (2004) defined the soluble volume fraction of the particle, i.e. the volume that would correspond to the specified salt, as $\varepsilon = (G_f^3 - 1)/(G_{f\,salt}^3 - 1)$. In a later work, Rissler et al. (2006) proposed the alternative use of the parameter $\kappa_R$, which represents the number of soluble moles of ions per particle dry volume unit, and is related to the soluble volume fraction by the expression $\kappa_R = \varepsilon \nu \rho_s / M_s$.





These two parameters can be converted to the effective hygroscopicity parameter $\kappa_p$ using the following relations suggested by Gunthe et al. (2009):

$$\kappa_p \approx \kappa_{p_{salt}} \varepsilon \approx \kappa_{P_{salt}} \kappa_R \frac{M_{salt}}{\upsilon_{salt} \rho_{salt}}$$ (B2)

Where the subscript *salt* refers to the reference salt used, $\kappa_{p_{salt}}$ is the hygroscopicity parameter determined for the salt and

5   $M$, $\rho$ and $\upsilon$ are the molar mass, density and dissociation factor of the salt, respectively.

**Appendix C: Additional definitions in the calculation of the CCN activation**

The (diffusional) growth coefficient for particles in Eq. (6) is defined (Pruppacher and Klett, 1997)

$$G = \left( \frac{\rho_w RT}{e_s D_v^* M_w} + \frac{L_e \rho_w}{k_a^* T} \left( \frac{L_e M_w}{RT} - 1 \right) \right)^{-1}$$ (C1)

where $e_s$ is the saturation vapor pressure, $L_e$ is the latent heat of evaporation the water vapor diffusivity in the air $D_v^*$ and

10   the thermal conductivity of air $k_a^*$ are corrected for non-continuous effects and depend on the droplet size:

$$D_v^* = \frac{D_v}{\dfrac{d}{d+2\Delta_v} + \dfrac{2D_v}{d\,\alpha_c} \left( \dfrac{2\pi M_w}{RT} \right)^{1/2}}$$ (C2)

$$k_a^* = \frac{k_a}{\dfrac{d}{d+2\Delta_T} + \dfrac{2k_a}{d\,\alpha_T \rho_a c_{pa}} \left( \dfrac{2\pi M_a}{RT} \right)^{1/2}}$$ (C3)

$c_{pa}$ is the specific heat of air, values for the vapor and temperature jumps were, respectively, $\Delta_v = 1.096 * 10^{-7}\,\text{m}$ and

$\Delta_T = 2.16 * 10^{-7}\,\text{m}$, and the condensation and thermal accommodation coefficient were chosen as $\alpha_c = 1.0$ and $\alpha_T = 0.96$.

15   Additionally, size-independent coefficients in Eq. (9) for the supersaturation rate of change are defined as (Seinfeld and Pandis, 2006),

$$\alpha = \frac{L_e M_w g}{c_{pa} R T^2} - \frac{g M_a}{RT}$$ (C4)

$$\gamma = \frac{p M_a}{e_s M_w} + \frac{M_w L_e^2}{c_{pa} R T^2}$$ (C5)





**Appendix D: Simplified Köhler equation and estimation of the cut diameter for CCN activation**

For an aerosol particle with dry diameter $d_{dry}$ and formed by a soluble fraction and an insoluble core, the Köhler equation can be approximated by the expression (Pruppacher and Klett, 1997):

$$S \approx 1 + \frac{A}{d} - \frac{B\,d_{dry}^3}{d^3 - d_{dry}^3} \tag{D1}$$

where the Kelvin and Raoult terms are estimated, respectively, as $A = \dfrac{4 M_w \sigma_w}{RT \rho_w}$ and $B = \dfrac{\nu \varepsilon M_w \rho_s}{M_s \rho_w}$, and all symbols are

described in the Appendix A. In this work, $B$ was assumed to be identical to the parameter $\kappa_P$ for all values of $\kappa_p$ and $S_c$.

It can be showed (Pruppacher and Klett, 1997) that the particle cut wet diameter for activation $d_c$ can be estimated as:

$$d_c = -D + \left(D^2 - E\right)^{1/2} \tag{D2}$$

where the parameters $D$ and $E$ are estimated as:

$$D = \frac{B^2 A - 3 B A s}{3 B s^2 - 3 B^2 s} \tag{D3}$$

and

$$E = \frac{3 B A^2}{3 B s^2 - 3 B^2 s} \tag{D4}$$

Finally, the corresponding dry diameter of the smallest activated particle, $d_{dry,c}$, can be calculated as:

$$d_{dry,c}^3 = \frac{d_c^3 (A - s\,d_c)}{A + (B - s) d_c} \tag{D5}$$

**Acknowledgements**

The São Paulo Research Foundation (FAPESP), through the projects 2012/13575-9, DR 2012/09934-3, BEPE 2013/02101-9 and BPE 2014/01564-8 supported this work. We also thank the PIs and staff of Balbina, Abracos_Hill and Porto_Velho_UNIR AERONET sites for their effort in establishing and maintaining the sites, in particular, during the periods discussed in this work. In addition, we thank Ricardo Almeida de Siqueira for reading the manuscript and giving valuable suggestions to improve its structure before submission.





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



**Table 1. Amazonian biomass burning number size distribution: 3 log-normal fits for CLAIRE, SMOCC and SAMBBA experiments.** $N_m$, $d_{g_m}$ and $\sigma_m$ **refer to the mode number concentration, geometric mean diameter and geometric standard deviation, respectively.**

| Experiment/mode | $N_m$ (cm⁻³) | $d_m$ (nm) | $\sigma_m$ | Notes and references |
|---|---|---|---|---|
| **CLAIRE** | | | | Balbina, Brazil, LBA-CLAIRE |
| *Recent smoke* | | | | 2001, wet-to-dry transition period |
| Nucleation | 302 | 14.0 | 1.31 | 2001 (Rissler et al., 2004). |
| Aitken | 280 | 69.0 | 1.35 | Recent smoke refers to an hours-old |
| Accumulation | 529 | 148.0 | 1.43 | biomass burning plume (dry crops |
| *Aged smoke* | | | | residues), duration 3 days. |
| Nucleation | 276 | 15.0 | 1.29 | The aged smoke period (duration 4 |
| Aitken | 304 | 68.0 | 1.32 | days) was considered to be |
| Accumulation | 736 | 139.0 | 1.45 | representative of 2.5-5 days aged smoke. |
| **SMOCC** | | | | Rondônia, Brazil, LBA-SMOCC |
| *Dry season*[a] | | | | 2002 (Rissler et al., 2006). |
| Nucleation | 1090 | 12.0 | 1.82 | Data can be considered |
| Aitken | 5213 | 92.0 | 1.63 | representative of regional haze in |
| Accumulation | 5214 | 190.0 | 1.53 | the region and includes both fresh |
| *Dry to wet period*[a] | | | | and aged BB aerosols. |
| Nucleation | 841 | 12.0 | 1.89 | Diurnal averages fits for the dry |
| Aitken | 984 | 66.0 | 1.39 | season and dry to wet transition |
| Accumulation | 3708 | 131.0 | 1.69 | periods are presented. |
| **SAMBBA** | | | | Porto Velho, Brazil, SAMBBA |
| Nucleation | 948 | 14.2 | 2.50 | 2012 (Brito et al., 2014). |
| Aitken | 4071 | 98.1 | 1.78 | Averages for the campaign, |
| Accumulation | 1063 | 179.1 | 1.48 | includes both fresh and aged BB aerosols. |





**Table 2. Effective hygroscopicity parameter $\kappa_{p,group}$ and aerosol fraction $f$ (number fraction times frequency of occurrence) for hygroscopic groups with very low hygroscopicity (VLH, $\kappa_p < 0.1$) and low hygroscopicity (LH, $0.1 \leq \kappa_p < 0.2$), and mode- and population effective $\kappa_{p_{eff}} = \sum \kappa_{p,group} AF_{group}$. Values are given for particles in the Aitken mode ($30\ \text{nm} < d_{dry} < 100\ \text{nm}$),**

5   **accumulation mode ($100\ \text{nm} \leq d_{dry} < 300\ \text{nm}$), and Aitken mode plus accumulation mode ($30\ \text{nm} \leq d_{dry} < 300\ \text{nm}$) dry sizes ranges.**

| *Period* | $\kappa_{p,VLH}\ /\ f$ | $\kappa_{p,LH}\ /\ f$ | $\kappa_{p_{eff}}$ | *Notes and references* |
|---|---|---|---|---|
| ***CLAIRE*** | | | | Balbina, Brazil, LBA-CLAIRE |
| *Recent smoke* | | | | wet-to-dry transition period 2001. |
| Aitken | 0.026 / 0.24 | 0.128 / 0.76 | 0.103 | |
| Accumulation | 0.052 / 0.15 | 0.182 / 0.85 | 0.163 | $\kappa$ values calculated from $\varepsilon$ |
| Aitken+Accumulation | 0.039 / 0.19 | 0.155 / 0.81 | 0.133 | values reported by Rissler et al. |
| *Aged smoke* | | | | (2004), where ammonium |
| Aitken | 0.017 / 0.33 | 0.139 / 0.67 | 0.096 | hydrogen sulfate was used to |
| Accumulation | 0.059 / 0.11 | 0.173 / 0.89 | 0.160 | represent the soluble fraction. |
| Aitken+Accumulation | 0.038 / 0.22 | 0.156 / 0.78 | 0.128 | |
| | | | | |
| ***SMOCC*** | | | | Rondônia, Brazil, LBA-SMOCC |
| *Afternoon Averages* | | | | 2002, during the dry season and |
| *Dry period* | | | | dry to wet transition periods. |
| Aitken | 0.051 / 0.90 | 0.146 / 0.10 | 0.061 | Afternoon averages (1200-1600 |
| Accumulation | 0.068 / 0.81 | 0.154 / 0.19 | 0.084 | local time) were calculated from |
| Aitken+Accumulation | 0.059 / 0.85 | 0.150 / 0.15 | 0.072 | $\kappa_R$ (Vestin et al., 2007) and daily |
| *Dry to wet period* | | | | averages were calculated from H- |
| Aitken | 0.061 / 0.72 | 0.154 / 0.28 | 0.087 | TDMA $G_f$ data (Rissler et al., |
| Accumulation | 0.064 / 0.5 | 0.172 / 0.5 | 0.119 | 2006). |
| Aitken+Accumulation | 0.062 / 0.61 | 0.163 / 0.39 | 0.103 | |
| *Diurnal averages* | | | | |
| *Dry period* | | | | |
| Aitken | 0.032 / 0.93 | 0.120 / 0.07 | 0.038 | |
| Accumulation | 0.041 / 0.80 | 0.119 / 0.20 | 0.056 | |
| Aitken+Accumulation | 0.037 / 0.86 | 0.119 / 0.14 | 0.048 | |
| *Dry to wet period* | | | | |
| Aitken | 0.038 / 0.87 | 0.131 / 0.13 | 0.050 | |
| Accumulation | 0.042 / 0.59 | 0.127 / 0.41 | 0.077 | |
| Aitken+Accumulation | 0.040 / 0.73 | 0.129 / 0.27 | 0.064 | |



**Table 3. Parameters for the Aitken and accumulation log-normal number size distribution for the defined case studies.**

| | $N_m$ (cm$^{-3}$) | $d_m$ (nm) | $\sigma_m$ |
|---|---|---|---|
| *Case MP$_{5,1}$* | | | |
| Aitken | 5000 | 95 | 1.60 |
| Accumulation | 1000 | 180 | 1.50 |
| *Case MP$_{1,5}$* | | | |
| Aitken | 1000 | 95 | 1.60 |
| Accumulation | 5000 | 180 | 1.50 |
| *Case HP$_{5,5}$* | | | |
| Aitken | 5000 | 95 | 1.60 |
| Accumulation | 5000 | 180 | 1.50 |





**Table 4. Number fractions for the hygroscopic groups in the externally mixed populations *Ext1* and *Ext2*.**

| $\kappa_{p_{eff}} = \sum \kappa_{p_{hg}} f_{hg}$ | Ext1 | | Ext2 | |
|---|---|---|---|---|
| | $f_{\kappa_p=0.04}$ | $f_{\kappa_p=0.16}$ | $f_{\kappa_p=0.04}$ | $f_{\kappa_p=0.30}$ |
| 0.04 | 1.00 | 0.00 | 1.00 | 0.00 |
| 0.06 | 0.83 | 0.17 | 0.92 | 0.08 |
| 0.08 | 0.67 | 0.33 | 0.85 | 0.15 |
| 0.10 | 0.50 | 0.50 | 0.77 | 0.23 |
| 0.12 | 0.33 | 0.67 | 0.69 | 0.31 |
| 0.14 | 0.17 | 0.83 | 0.62 | 0.38 |
| 0.16 | 0.00 | 1.00 | 0.54 | 0.46 |
| 0.18 | - | - | 0.46 | 0.54 |
| 0.20 | - | - | 0.38 | 0.62 |
| 0.25 | - | - | 0.19 | 0.81 |
| 0.30 | - | - | 0.00 | 1.00 |





**Table 5. Parameters for the simulations.**

| Parameter | Value / Range |
|---|---|
| Updraft velocity | 0.1 - 10 m s$^{-1}$ |
| Hygroscopicity parameter | |
| *Int* | 0.02 - 0.60 |
| *Ext1* | 0.04 - 0.16 |
| *Ext2* | 0.04 - 0.30 |
| Initial conditions | |
| Relative humidity | 98 % |
| Temperature | 93 K |
| Atmospheric pressure | 900 hPa |
| Air parcel height | 500 m |







**Figure 1. Precipitation (mm/day) (color scale, left), precipitation anomaly (mm/day) (color scale, right) and wind circulation at 850 hPa level (streamlines, all) during CLAIRE 2001 (a, b), SMOCC 2002 DS (c, d) and TP (e, f) periods, and SAMBBA 2012 (g, h). Open circles denote the corresponding experiment ground site. Open triangles indicate, respectively, Balbina (a, b), Abracos_Hill (c, d, e, f) and Porto_Velho_UNIR (g, h) AERONET stations.**



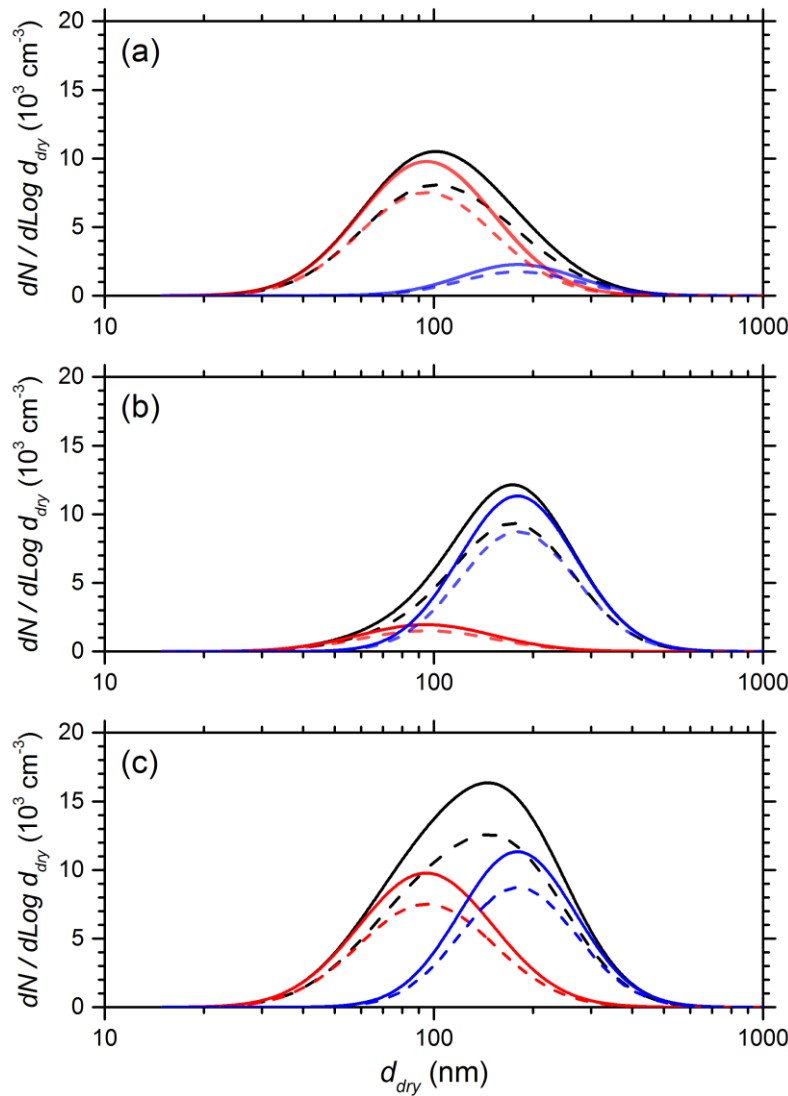

**Figure 2.** Schematic number size distributions for MP$_{5,1}$ (a), MP$_{1,5}$ (b) and HP$_{5,5}$ (c) case studies. Total population (black, solid), Aitken (red, solid) and accumulation (blue, solid) modes are indicated. Particles in hygroscopic group $\kappa_p = 0.04$ (dashed line, all colors) are also showed for a population average $\kappa_p = 0.10$.



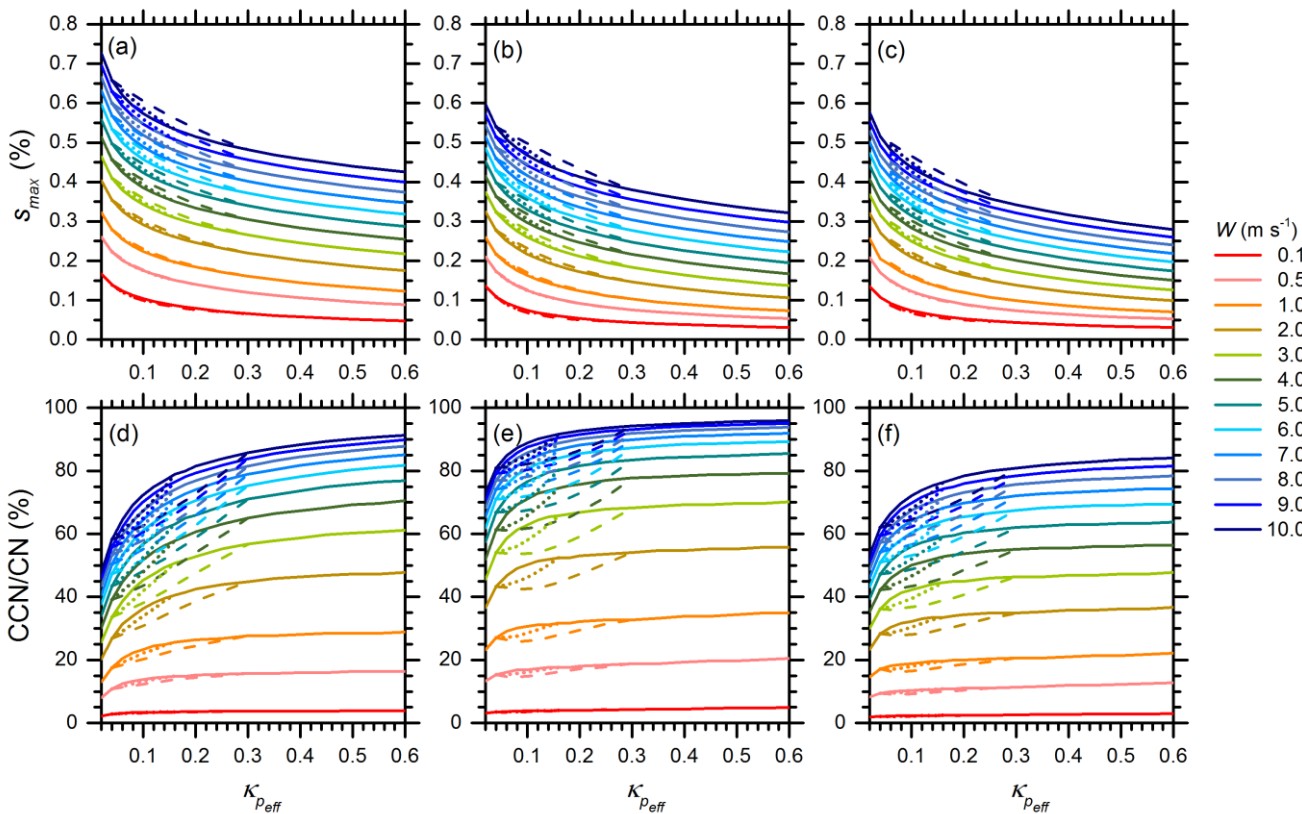

**Figure 3. Maximum supersaturation reached (top) and fraction of particles activated as CCN (bottom) for the internal mixing (solid line) and external mixing cases *Ext1* (dotted line) and *Ext2* (dashed line). Plots on columns (a, d), (b, e) and (c, e) are for MP$_{5,1}$, MP$_{1,5}$ and HP$_{5,5}$ case studies, respectively. The color scale refers to the updraft velocities from 0.1 m s$^{-1}$ and 10 m s$^{-1}$.**





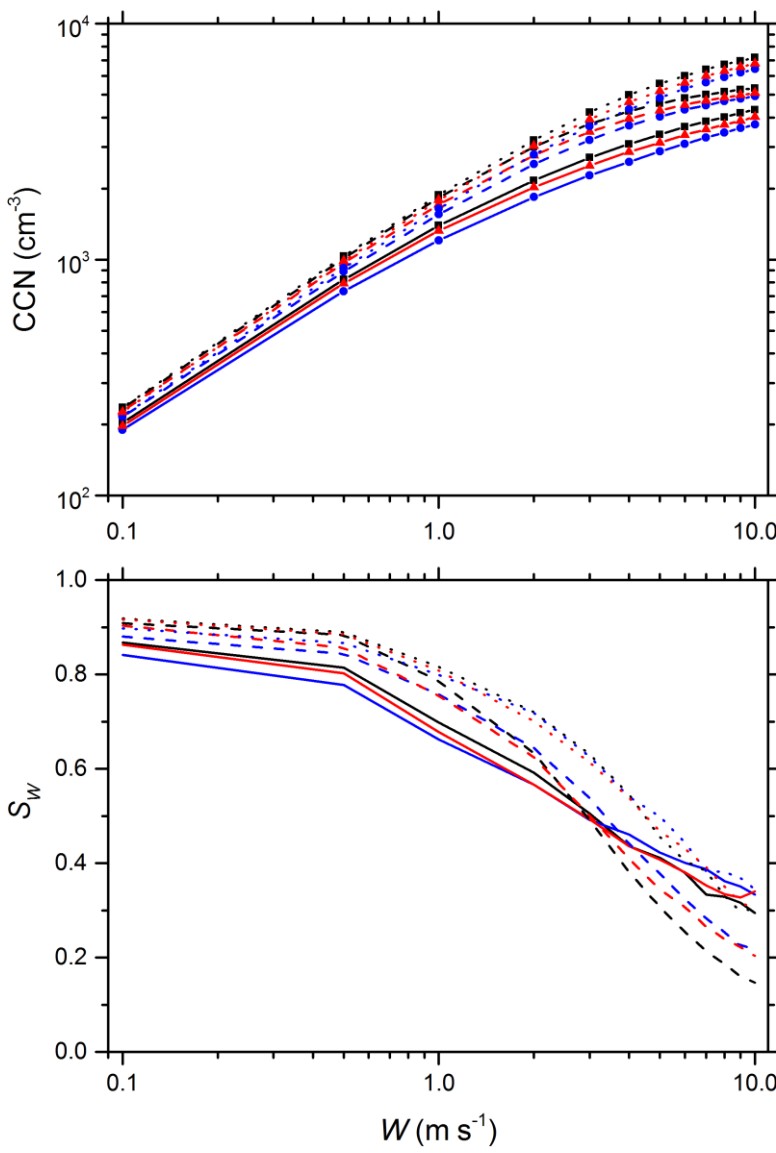

**Figure 4. Number of particles activated as CCN (top) and sensitivity $S_W$ of CCN to the updraft velocity $W$ (bottom) for $\kappa_p = 0.10$, obtained for the MP$_{5,1}$ (solid line), MP$_{1,5}$ (dashed line) and HP$_{5,5}$ (dotted line) case studies. Results for internal mixed *Int* population and externally mixed populations *Ext1* and *Ext2* are in black, red and blue, respectively.**



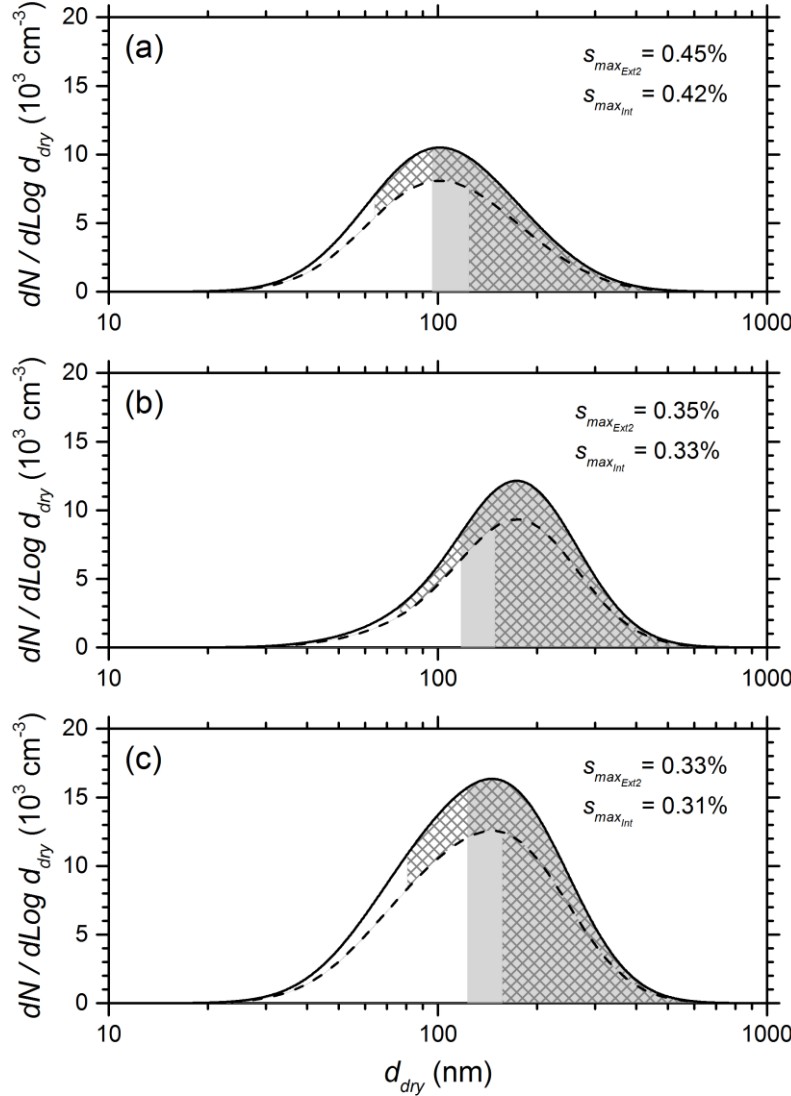

**Figure 5. Schematic number size distribution of particles activated as CCN in *Ext2* (angled grid area) and *Int* (grey area) mixing states, for an average $\kappa_p = 0.1$ and $W = 5$ m s$^{-1}$, for (a) MP$_{5,1}$, (b) MP$_{1,5}$ and (c) HP$_{5,5}$ case studies. Total aerosol population (black, solid line), hygroscopic group $\kappa_p = 0.04$ (black, dashed line) and maximum supersaturation reached in the simulations for each mixing state are indicated.**




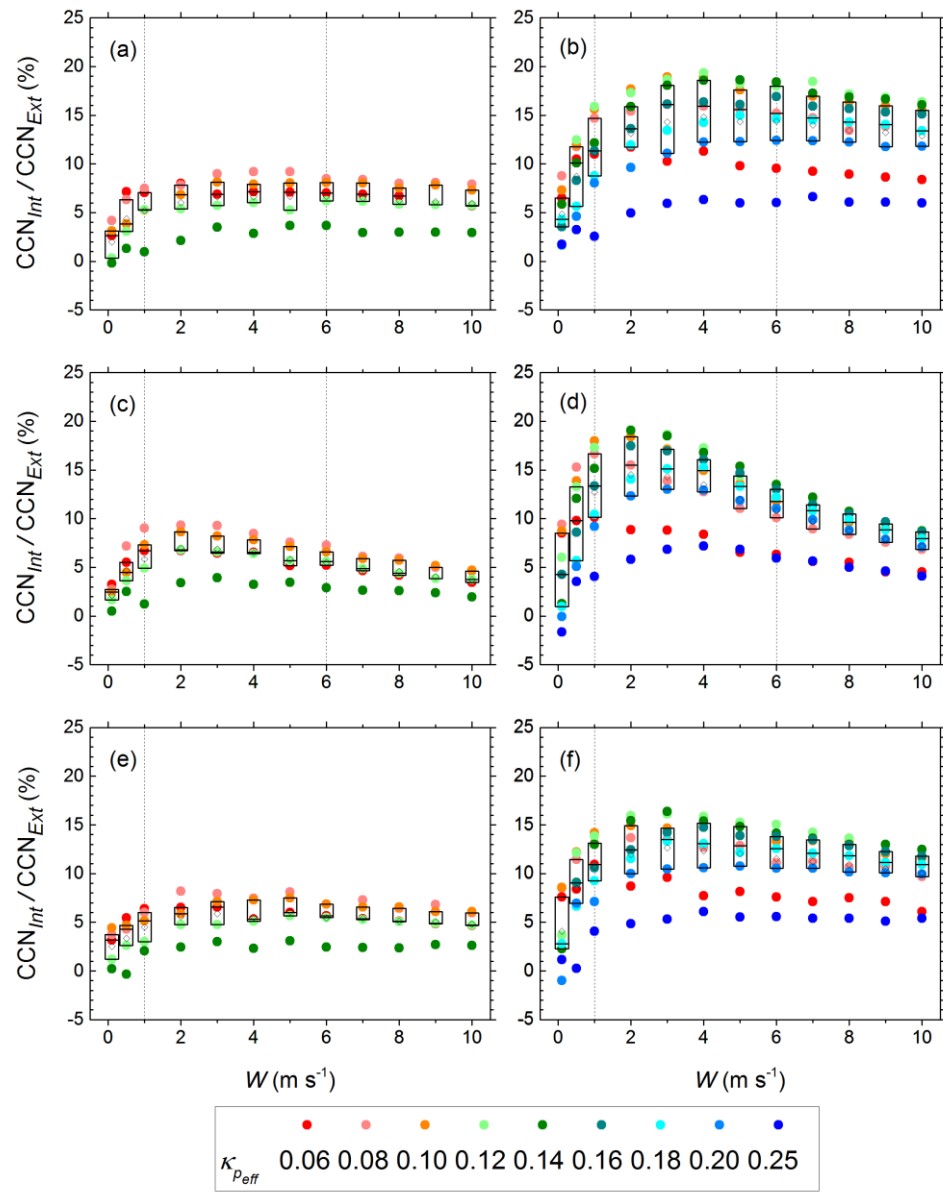

**Figure 6. CCN overestimation when the aerosol is assumed to internally mixed, calculated as a function of the hygroscopicity (color scale) and the updraft velocity, for the external mixing _Ext 1_ (left) and _Ext 2_ (right). Plots on panels (a, b), (c, d) and (e, f) correspond to MP$_{5,1}$, MP$_{1,5}$ and HP$_{5,5}$ case studies, respectively. Box plots on top of data represent the spread for different hygroscopicity parameters. The box boundaries delimitate the interquartile range and mean values are indicated by diamond symbols. Dashed lines represent the approximate boundaries between CCN activation regimes.**



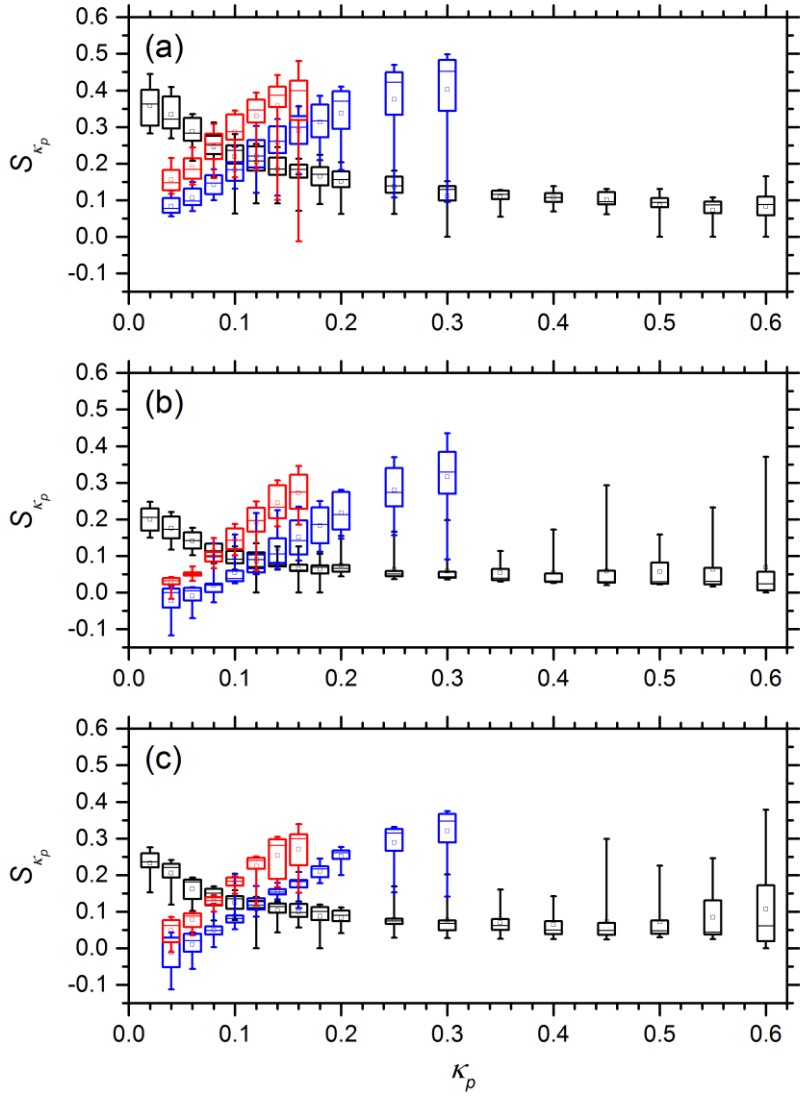

**Figure 7. Box-whisker plots of the sensitivity** $S_{\kappa_p}$ **of CCN activation to the hygroscopicity parameter** $\kappa_p$, **showing spread of results for updraft velocities between 0.1 m s$^{-1}$ and 10 m s$^{-1}$, for (a) MP$_{5,1}$, (b) MP$_{1,5}$ and (c) HP$_{5,5}$ case studies. Box bounds show the interquartile range, the mean value is indicated by a small square and whiskers delimitate minimum and maximum values. Results for the internally mixed *Int* and externally mixed populations *Ext1* and *Ext2* are plotted in black, red and blue, respectively.**





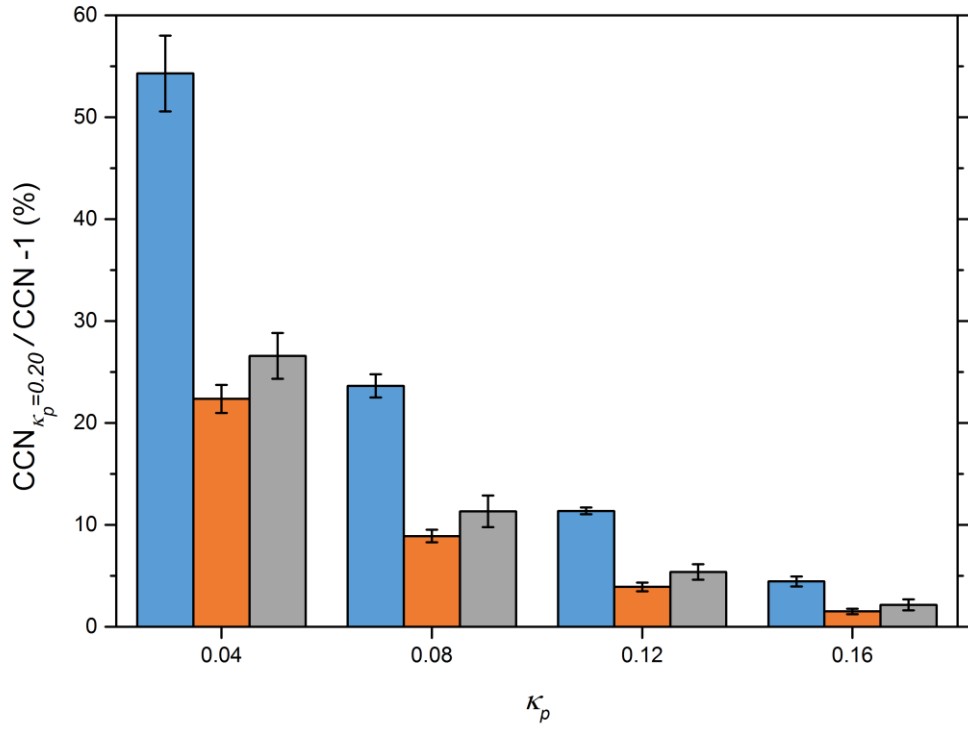

**Figure 8. Overestimation of the CCN activation (mean ± standard deviation over the updraft velocities in the updraft- and aerosol sensitive regime) when $\kappa_p = 0.20$ is assumed, as a function of the population $\kappa_p$. Results correspond to MP$_{5,1}$ (blue), MP$_{1,5}$ (orange) and HP$_{5,5}$ (grey) case studies for an internally mixed population.**





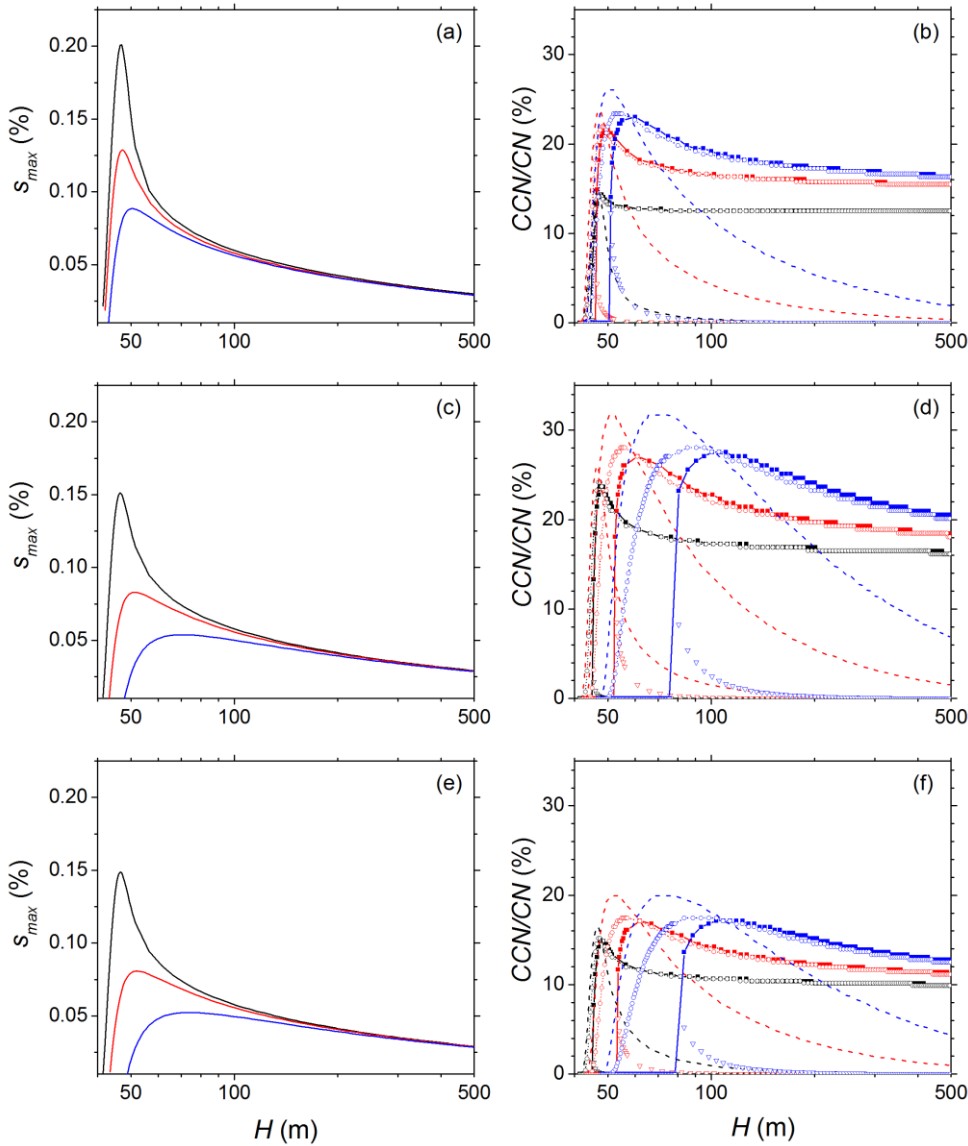

**Figure 9 Supersaturation (left) and CCN/CN (right) as a function of cloud height for an internally mixed population with** $\kappa_p = 0.06$ **(black),** $\kappa_p = 0.25$ **(red) and** $\kappa_p = 0.60$ **(blue), and** $W = 0.5$ m s$^{-1}$. **The cloud droplet concentration was estimated either as** $\mathrm{CCN}_{eq}$ **(dashed line),** $\mathrm{CCN}_{neq,simp}$ **(solid line, open circles) or** $\mathrm{CCN}_{neq}$ **(solid line, close squares). The fraction of the population not strictly activated in** $\mathrm{CCN}_{neq}$ **is indicated (open down triangles). Plots on panels (a, b), (c, d) and (e, f) correspond to MP$_{5,1}$, MP$_{1,5}$ and HP$_{5,5}$ case studies, respectively.**



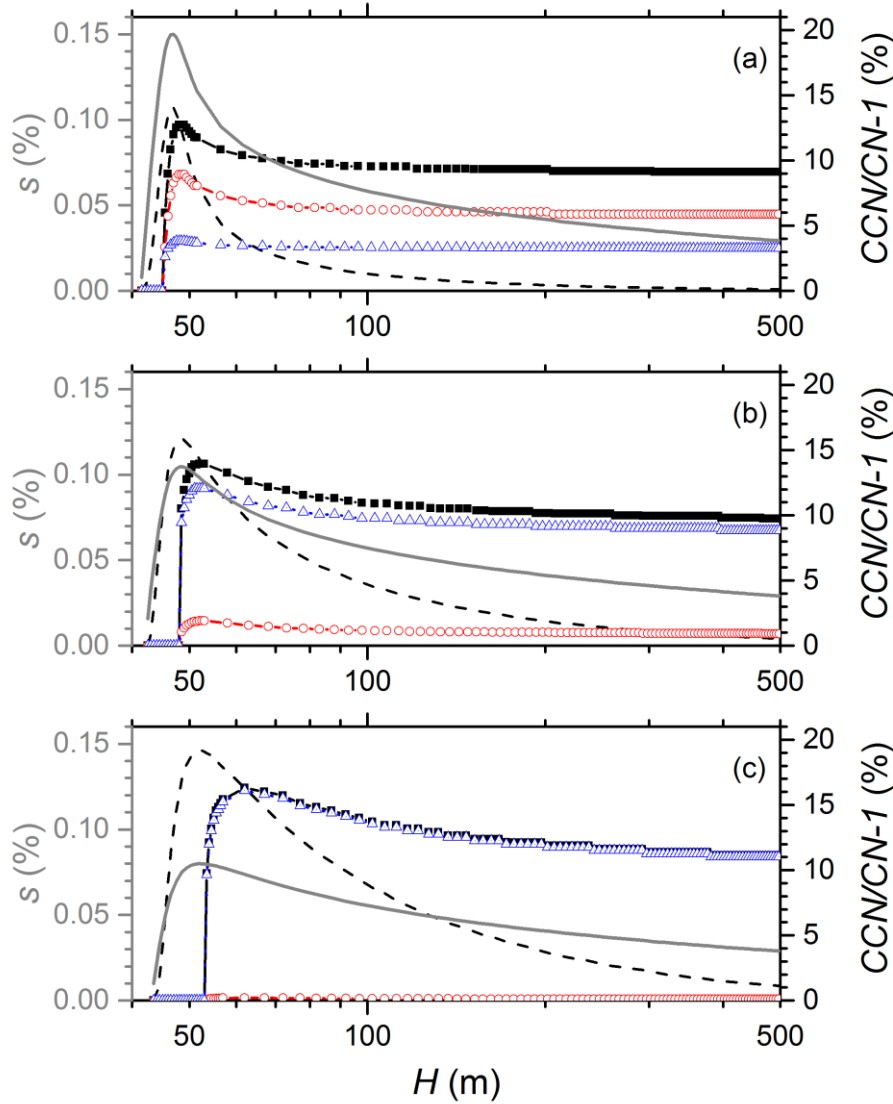

**Figure 10 Supersaturation (left axis, grey) and CCN/CN during the simulation (right axis) for the *Ext2* population and the HP$_{5,5}$ case study, for $W$ = 0.5 m s$^{-1}$ and $\kappa_{p_{eff}} = 0.06$ (a), $\kappa_{p_{eff}} = 0.14$ (b) and $\kappa_{p_{eff}} = 0.25$ (c). The cloud droplet concentration was estimated as $CCN_{eq}$ (dashed line), and $CCN_{neq}$ for the population (black solid line, close squares) and hygroscopic groups with $\kappa_p = 0.04$ (red dashed line, open circles) and $\kappa_p = 0.30$ (blue dotted line, open up triangles).**





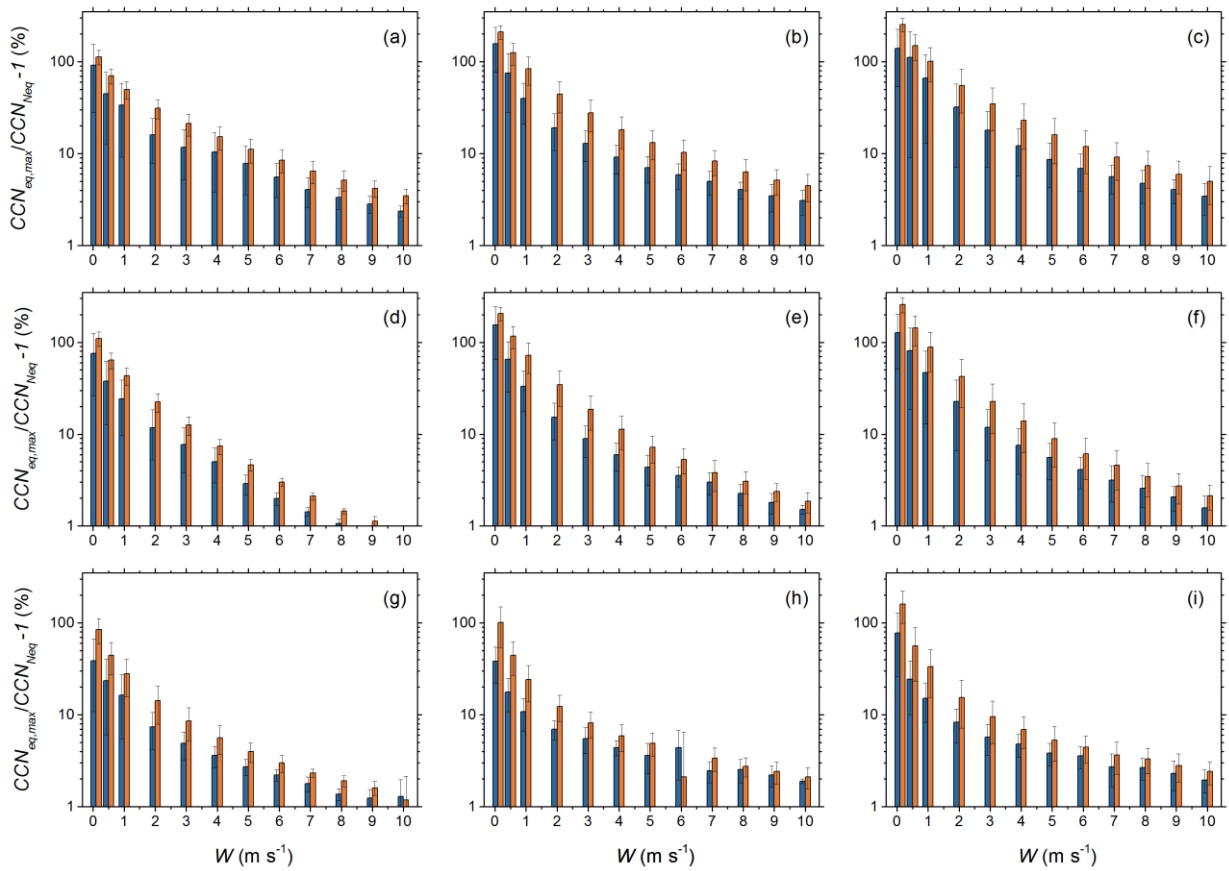

**Figure 11 Overestimation when the CCN population is estimated assuming equilibrium at the time of maximum supersaturation ($CCN_{eq}$), compared with $CCN_{neq}$ at the time of maximum supersaturation and at the end of the simulation, for the range of updraft velocities. Values correspond to the MP$_{5,1}$ (a, b and c panels), MP$_{1,5}$ (d, e and f panels) and HP$_{5,5}$ (g, h and i panels) case studies. The mixture of the aerosol population was either internal (left panels), or external as in *Ext1* (middle panels) and *Ext2* (right panels).**