# Peer review of "Impact of mixing state and hygroscopicity on CCN activity of biomass burning aerosol in Amazonia"

_Atmospheric Chemistry and Physics, 2016_

## Referee Comment (RC1) · Anonymous Referee #1 · 23 Jun 2016

Study by Gacita et al investigate hygroscopicity and mixing state influence on CCN activity of biomass burning aerosol in Amazonia using own adiabatic rising air parcel cloud model. Main aim is to assess effect of using various parametrizations of biomass burning aerosols in modelling CCN activation.

Main results of this study are: 1) use of the Kp for continental and biomass burning aerosol elsewhere can result in large overestimation of CCN over Amazonia. 2) Mixing state assumptions play less significant role 3) Kinetic limitations are not important. But they are lost in a long text full of detail information. On my opinion the manuscript can be reduce to a "letter" type of article.

Overall it is a valuable study on an important subject, but in a form as it is presented, it

is a numerical exercise based on synthetic input loosely linked to observations and with limited impact and link to reality. Study on nearly the same subject done by Roberts et al (JGR, Vol 108, 2003 doi:10.1029/2001JD000985) is not used and referenced at all and it can provide good observational and modelling basis for the sensitivity study in current manuscript, especially with respect to uncertainty, variability and error analysis.

I cannot recommend the manuscript for publication in its current form and would like to encourage authors towards a better manuscript appropriate for publication in ACP.

Detail comments:

Chapters 2.1 -2.3 covers summary of basic textbook equations reported in numerous publications in past. I suggest to move these chapters to Appendix or Supplementary material and reduce it with proper references to paragraph or two in paper itself.

Chapter 3 should be reduced significantly. It is not aim of this paper to make an overview of the past experiments. Data from each experiment used in this study can be properly referenced and briefly described in one paragraph. Chapter 3.1 is irrelevant for this study and should be removed completely. Chapter 3.2 should be significantly reduced and combined with paragraphs describing individual experiments, which pro-vided observational basis for this study.

P1L23: why original reference to Köhler paper from 1936 is not included?

P15 L24-26: underestimation with respect to what? External mixing state?

P16 L1: overestimation with respect to....?

P16 L15-25: How close to reality are selected externally and internally mixed fractions? It is not clear to me if it is based on observational evidence or just assumed for test purposes.

---

## Referee Comment (RC2) · Anonymous Referee #2 · 29 Jun 2016

The paper presents a sensitivity study of parcel-model simulated droplet number to mixing-state and potential kinetic limitation to droplet growth. The study is extensively stated as being motivated by a number of S. American biomass burning experiments and the model parameter range inspired, though not tightly constrained, by observations from such experiments. I think there are some interesting results that are potentially worthy of publication - the unimportance of kinetic limitation and the roles that assumptions of mixing state and of kappa based on non-Amazonian observations are of note. However, I don't think they are done very good justice by the current form of the paper, which I found rather meandering and over-long. I found the abstract to be too discursive and not terribly useful in not providing a quantitative precis of the key

findings to make it easy for a reader to digest the study. I similarly found the main messages were buried in long discursive sections throughout the paper. A more tightly focussed presentation of the key results, conclusions and recommendations is needed to bring out the novelty and significance of the study. There are also a few places where the paper seems to be imprecise and appear to convey a lack of understanding that should be clarified (see specific points below). I won't try to rewrite the paper, but sections 3 and 4 are very long and could conceivably be replaced by a couple of paragraphs referencing tables 1 to 5 to state exactly how measurements were used to define the parameter range for the model sensitivities, simply referencing the relevant observational publications. Such a concise presentation would better allow a reader to gauge exactly how much observational constraint is provided for the sensitivity analysis without having to dig out the material. Furthermore, I would expect the manuscript to provide some recommendation for how the findings may be able to inform the treatments in regional coupled models, general circulation models or earth system models, given the diversity of representations of size and composition resolved aerosol and parameterisations of droplet activation. Some model treatments (e.g. the M7, GLOMAP or MOSAIC aerosol variants with Abdul-Razzak and Ghan, Fountoukis and Nenes or Barahona et al. activation parameterisations) are reasonably close to being able to capture the effects mentioned in the paper and do not make such coarse approximations as the base case assumptions, so it is not clear which models will have problems of the magnitude identified. Indeed it is unclear whether such a scale of uncertainty is significant given the other sub-grid difficulties such as representation of updraughts. Some discussion of these aspects should be included at the expense of concision in the more superfluous material.

Below I've listed a few specific areas which the authors should also pay attention to in any revised submission.

Figure 1 is unnecessary to the paper, providing a bit of background context and motivation that can be found elsewhere. At most it is supplementary material or appropriate

for an appendix. If it were to remain, I would expect a model sensitivity study to look at the sensitivity of precipitation to mixing state. This would need a much more sophisticated model than used in the current paper.

Sections 2.1 to 2.3 do not present any new approaches and can be replaced by a much shorter section, relegating the rest to the Appendix or to supplementary material or simply referenced. However, some of the things they don't cover appear quite relevant to the study and should be addressed by the authors: i) there can be a strong sensitivity of predicted droplet number to the initial conditions, in particular the height at which an aerosol population is assumed to be in equilibrium with the ambient RH. Table 5 states that the parcel is initiated at 98% RH. Presumably the aerosol populations are assumed to be at equilibrium here. This RH is very close to cloudbase. A mixture of different hygroscopicity of particles will have very different masses of associated water and may have competed for available water more or less successfully already by this stage and may not be at their equilibrium size, dependent on the number of particles in the population. The dependence on initialisation conditions (80, 85, 90, 95, 98, 99% RH, for example) for different updraughts and size distributions may be particularly important for externally-mixed populations. The authors need to demonstrate that 98% is a justifiable initialisation for the entire range of updraughts and particle distributions in their study. ii) the surface tension of water dependence on temperature may be of some modest importance as Christensen and Petters claim. However, the current manuscript completely ignores the very extensive literature on the roles of surface tension and bulk-to-surface partitioning that has been backwards and forwards in the literature since 1999. This is particularly relevant for particles heavily dominated by the organic components present during biomass burning. The authors need to justify ignoring any discussion or treatment of this, particularly given the recent claims of the pendulum swinging back towards an extremely strong enhancement of activation of organic-rich particles. I do not necessarily expect the authors to agree with these claims, but think it must be considered a first order sensitivity in BB dominated aerosol to be folded into the mixing-state and kinetic limitation discussion.

p7 line 11, it is incorrect to state that "McFiggans et al. (2006) proposed sensitivities of the drop number concentration (CCN)..." and then state equation 7. They did propose the method to state sensitivities, but did so with cloud droplet number ($N_d$). Clearly CCN are not droplets. This sentence can simply be rephrased, but the implications of the underlying understanding of the problem are worrying.

If the authors were to provide a tighter and more focussed version of the manuscript addressing the points above, I think it would be a useful contribution to the literature.

---

## Author Comment (AC1) · 11 Aug 2016

Response to referee #1

We thank the referee #1 for the careful review of the manuscript and for providing helpful comments on how it could be improved. General comments of referee #1 on the article's form are accepted and will be considered in a manuscript to be submitted for reconsideration. Replies to specific comments and questions raised can be found below.

"Study on nearly the same subject done by Roberts et al (JGR, Vol 108, 2003 doi:10.1029/2001JD000985) is not used and referenced at all and it can provide good

observational and modelling basis for the sensitivity study in current manuscript, especially with respect to uncertainty, variability and error analysis."

Roberts el at (2003) should indeed be referenced as an important precedent to this work and this will be corrected in the manuscript to be submitted for reconsideration. In particular, it was interesting to note that Roberts et al found also discussed the effect of kinetic limitations, reporting a reduction in the droplet concentrations of up to 35% for the dry season and updrafts of 0.1 ms-1, when comparing with the value estimated assuming equilibrium Köhler theory. Our results for low hygroscopicity values agree reasonably with this estimation, considering the differences between both studies (specially, size distribution and the representation of the aerosol), even when we found the overestimation to be much larger (up to $\sim$100% for internal mixings and up to $\sim$250% for external ones) for larger values of hygroscopicity. We thank referee #1 for pointing this out since it will certainly enrich the discussion.

"Detail comments: Chapters 2.1 -2.3 covers summary of basic textbook equations reported in numerous publications in past. I suggest to move these chapters to Appendix or Supplementary material and reduce it with proper references to paragraph or two in paper itself. Chapter 3 should be reduced significantly. It is not aim of this paper to make an overview of the past experiments. Data from each experiment used in this study can be properly referenced and briefly described in one paragraph. Chapter 3.1 is irrelevant for this study and should be removed completely. Chapter 3.2 should be significantly reduced and combined with paragraphs describing individual experiments, which provided observational basis for this study."

The manuscript to be submitted for reconsideration will be modified accordingly.

"P1L23: why original reference to Köhler paper from 1936 is not included?"

This will be corrected in the in the manuscript to be submitted for reconsideration.

"P15 L24-26: underestimation with respect to what? External mixing state? P16 L1:

overestimation with respect to: : :.?"

In both Ext1 and Ext2 situations, it is assumed that the aerosol particles are externally mixed. Therefore, the external mixing is the reference case. Assuming internal mixing typically leaded to an underestimation of the maximum supersaturation reached, and to an overestimation of the aerosol activated fraction. The sections when this is not clearly specified will be corrected.

"P16 L15-25: How close to reality are selected externally and internally mixed fractions? It is not clear to me if it is based on observational evidence or just assumed for test purposes."

This specific case was selected to illustrate graphically the impact of mixing state. Observational data for Amazon biomass burning is better described by the Ext1 externally mixed population, and the impact of mixing state in Ext1 was much lower than what is showed in figure 5, with average overestimations below 6% (P17 L8-18).

Response to referee #2

We thank the referee #2 for the careful review of the manuscript and for providing helpful comments on how it could be improved. General comments of referee #2 on the article's form are accepted and will be considered in a manuscript to be submitted for reconsideration. Replies to specific comments and questions raised can be found below.

"Furthermore, I would expect the manuscript to provide some recommendation for how the findings may be able to inform the treatments in regional coupled models, general circulation models or earth system models, given the diversity of representations of size and composition resolved aerosol and parameterisations of droplet activation. Some model treatments (e.g. the M7, GLOMAP or MOSAIC aerosol variants with Abdul-Razzak and Ghan, Fountoukis and Nenes or Barahona et al. activation parameterisations) are reasonably close to being able to capture the effects mentioned in the

paper and do not make such coarse approximations as the base case assumptions, so it is not clear which models will have problems of the magnitude identified."

We agree with referee #2 in that models are able to capture the effects of hygroscopicity and internal/external mixing state. Most of them also can consider to some degree the impact of kinetic limitations, although variants of Abdul-Razzak and Ghan parameterizations are widely used and do not consider these effects. The choice of to use two separate aerosol populations to account for the externally mixing character of the biomass burning population will increase the computational burden of the model. The modeler might choose instead to consider biomass burning aerosols as only one population internally mixed and externally mixed with other aerosol populations, unless given sufficient evidence that the overestimation derived from this choice is significant (and, in the case of amazon biomass burning aerosols, it seems that it is not). In a similar way, most global models or regional models over a large domain can allow for the specification of the aerosol hygroscopicity for different regions, but it is much simpler to choose a single value for all biomass burning. The choice of a parameterization that accounts for kinetic limitations, typically more demanding in terms of computational resources, needs to be similarly justified. Thus, our work did not aim to suggest improvements of the parameterizations themselves, but rather to guide the modeler choices. This topic will be further explored in the manuscript to be resubmitted.

"Indeed it is unclear whether such a scale of uncertainty is significant given the other sub-grid difficulties such as representation of updraughts."

We agree with referee #2 in that there are another number of factors that also increase the level of uncertainties. Yet, to improve the representation of the aerosol processes in GCMs is of great importance to adequately simulate aerosol-cloud interactions and their impact in the climatic system. In this case, the suggestions for the modeling of biomass burning aerosols that arrive from our work are, for the most part, easy to implement, without requiring improvements in the existing parameterizations.

[Figure]

"Figure 1 is unnecessary to the paper, providing a bit of background context and motivation that can be found elsewhere. At most it is supplementary material or appropriate for an appendix. If it were to remain, I would expect a model sensitivity study to look at the sensitivity of precipitation to mixing state. This would need a much more sophisticated model than used in the current paper.

"Sections 2.1 to 2.3 do not present any new approaches and can be replaced by a much shorter section, relegating the rest to the Appendix or to supplementary material or simply referenced."

The manuscript to be submitted for reconsideration will be modified accordingly.

Specific points:

"i) there can be a strong sensitivity of predicted droplet number to the initial conditions, in particular the height at which an aerosol population is assumed to be in equilibrium with the ambient RH. Table 5 states that the parcel is initiated at 98% RH. Presumably the aerosol populations are assumed to be at equilibrium here. This RH is very close to cloudbase. A mixture of different hygroscopicity of particles will have very different masses of associated water and may have competed for available water more or less successfully already by this stage and may not be at their equilibrium size, dependent on the number of particles in the population. The dependence on initialisation conditions (80, 85, 90, 95, 98, 99% RH, for example) for different updraughts and size distributions may be particularly important for externally-mixed populations. The authors need to demonstrate that 98% is a justifiable initialisation for the entire range of updraughts and particle distributions in their study."

We thank referee #2 for raising this concern, and will discuss briefly this choice in the article to be submitted. We found that the influence of the initial relative humidity was very low. To illustrate this, the maximum supersaturation and aerosol activated fraction are shown in Figure 1 for a range of initial values of the relative humidity. The values were calculated for the three size distributions considered, considering the externally

mixed case Ext2 with a population average hygroscopicity of 0.10 and three values of updraft velocities, including the minimum and maximum values considered. We found only a weak dependence (differences between maximum supersaturations obtained initializing at 80% and at 99% below 0.03) of maximum supersaturations with the initial relative humidity for the highest updraft values, and a negligible effect in the activated fraction.

"ii) the surface tension of water dependence on temperature may be of some modest importance as Christensen and Petters claim. However, the current manuscript completely ignores the very extensive literature on the roles of surface tension and bulk-to-surface partitioning that has been backwards and forwards in the literature since 1999. This is particularly relevant for particles heavily dominated by the organic components present during biomass burning. The authors need to justify ignoring any discussion or treatment of this, particularly given the recent claims of the pendulum swinging back towards an extremely strong enhancement of activation of organic-rich particles."

This is an interesting point, and there is, as referee#2 points out, extensive literature on the topic including laboratory data specific for biomass burning that suggest this could be indeed an important issue (Fors et al., 2010; Giordano et al., 2013). However, it was not within the proposed scope of the submitted manuscript to approach this question, considering both the complexity of the biomass burning particles aerosol particles in terms of organic composition, and the scarcity of data to estimate this effects using, for instance, the methodology proposed by (Petters and Kreidenweis, 2013). We will acknowledge this limitation of the study within the article.

"p7 line 11, it is incorrect to state that "McFiggans et al. (2006) proposed sensitivities of the drop number concentration (CCN)..." and then state equation 7. They did propose the method to state sensitivities, but did so with cloud droplet number ($N_d$). Clearly CCN are not droplets. This sentence can simply be rephrased, but the implications of the underlying understanding of the problem are worrying."

[Figure]

We thank referee #2 for noting this. The expression will be rephrased.

Caption of Figure 1. Maximum supersaturation (top) and fraction of particles activated as CCN (bottom), as function of the initial relative humidity, for the MP5,1 (solid line, squares), MP1,5 (dashed line, circles), and HP5,5 (dotted line, triangles) size distributions and external mixing case Ext2 with a population average k_p=0.10. Values refer to updraft velocities W=0.1 m s-1 (red), W=3 m s-1 (green) and W=0.1 m s-1 (blue).

References:

Fors, E. O., Rissler, J., Massling, A., Svenningsson, B., Andreae, M. O., Dusek, U., Frank, G. P., Hoffer, A., Bilde, M., Kiss, G., Janitsek, S., Henning, S., Facchini, M. C., Decesari, S. and Swietlicki, E.: Hygroscopic properties of Amazonian biomass burning and European background HULIS and investigation of their effects on surface tension with two models linking H-TDMA to CCNC data, Atmos. Chem. Phys., 10(12), 5625–5639, doi:10.5194/acp-10-5625-2010, 2010.

Giordano, M. R., Short, D. Z., Hosseini, S., Lichtenmerg, W. and Asa-Awuku, A. A.: Changes in droplet surface tension affect the observed hygroscopicity of photochemically aged biomass burning aerosol., Environ. Sci. Technol., 47(3), 10980–10986, doi:10.1021/es404971u, 2013.

Petters, M. D. and Kreidenweis, S. M.: A single parameter representation of hygroscopic growth and cloud condensation nucleus activity – Part 3: Including surfactant partitioning, Atmos. Chem. Phys., 13(2), 1081–1091, doi:10.5194/acp-13-1081-2013, 2013.

[Figure]

**Fig. 1.**

---

## Author Response (AR1)

**Response to referee #1**

5    We thank the referee #1 for the careful review of the manuscript and for providing helpful comments on how it could be improved. General comments of referee #1 on the article's form are accepted and were considered in the manuscript submitted for reconsideration. Replies to specific comments and questions raised can be found below.

*"Study on nearly the same subject done by Roberts et al (JGR, Vol 108, 2003 doi:10.1029/2001JD000985) is not used and referenced at all and it can provide good observational and modelling basis for the sensitivity study in current manuscript,*

10    *especially with respect to uncertainty, variability and error analysis."*

Roberts el at (2003) should indeed be referenced as an important precedent to this work and this was corrected in the manuscript submitted both as a precedent work in the Introduction and within the discussion of kinetic limitations.

*"Detail comments:*

*Chapters 2.1 -2.3 covers summary of basic textbook equations reported in numerous publications in past. I suggest to move*

15    *these chapters to Appendix or Supplementary material and reduce it with proper references to paragraph or two in paper itself.*

*Chapter 3 should be reduced significantly. It is not aim of this paper to make an overview of the past experiments. Data from each experiment used in this study can be properly referenced and briefly described in one paragraph. Chapter 3.1 is irrelevant for this study and should be removed completely. Chapter 3.2 should be significantly reduced and combined with*

20    *paragraphs describing individual experiments, which provided observational basis for this study."*

The manuscript submitted for reconsideration was modified accordingly. Section 2 was reduced. Tables 1 and 2 were moved to supplementary material and section 3 and 4 were merged and reduced. Two appendices were removed.

*"P1L23: why original reference to Köhler paper from 1936 is not included?"*

This was corrected in the in the manuscript.

25    *"P15 L24-26: underestimation with respect to what? External mixing state?*

*P16 L1: overestimation with respect to: : :.?"*

In both Ext1 and Ext2 situations, it is assumed that the aerosol particles are externally mixed. Therefore, the external mixing is the reference case. Assuming internal mixing typically leaded to an underestimation of the maximum supersaturation reached, and to an overestimation of the aerosol activated fraction. The sections when this was not clearly specified were

30    modified accordingly.

*"P16 L15-25: How close to reality are selected externally and internally mixed fractions? It is not clear to me if it is based on observational evidence or just assumed for test purposes."*

This specific case was selected to illustrate graphically the impact of mixing state and this information was added to the manuscript. Observational data for Amazon biomass burning is better described by the Ext1 externally mixed population,

and the impact of mixing state in Ext1 was much lower than what is showed in figure 5, with average overestimations below 6% (P17 L8-18).

**Response to referee #2**

We thank the referee #2 for the careful review of the manuscript and for providing helpful comments on how it could be improved. General comments of referee #2 on the article's form were accepted and considered in the manuscript submitted for reconsideration. Replies to specific comments and questions raised can be found below.

*"Furthermore, I would expect the manuscript to provide some recommendation for how the findings may be able to inform the treatments in regional coupled models, general circulation models or earth system models, given the diversity of representations of size and composition resolved aerosol and parameterisations of droplet activation. Some model treatments (e.g. the M7, GLOMAP or MOSAIC aerosol variants with Abdul-Razzak and Ghan, Fountoukis and Nenes or Barahona et al. activation parameterisations) are reasonably close to being able to capture the effects mentioned in the paper and do not make such coarse approximations as the base case assumptions, so it is not clear which models will have problems of the magnitude identified."*

We agree with referee #2 in that models are able to capture the effects of hygroscopicity and internal/external mixing state. Most of them also can consider to some degree the impact of kinetic limitations, with variations of Abdul-Razzak and Ghan being a notable exception to this. The choice of to use two separate aerosol populations to account for the externally mixing character of the biomass burning population will increase the computational burden of the model. The modeler might choose instead to consider biomass burning aerosols as only one population internally mixed and externally mixed with other aerosol populations, unless given sufficient evidence that the overestimation derived from this choice is significant (which is the case of amazon biomass burning aerosols, is not). In a similar way, most global models or regional models over a large domain can allow for the specification of the aerosol hygroscopicity for different regions, but it is much simpler to choose a single value for all biomass burning. The choice of a parameterization that accounts for kinetic limitations, typically more demanding in terms of computational resources, needs to be similarly justified. Thus, our work did not aim to suggest improvements of the parameterizations themselves, but rather to guide the modeler choices. This topic was included in the conclusion, in the manuscript resubmitted.

*"Indeed it is unclear whether such a scale of uncertainty is significant given the other sub-grid difficulties such as representation of updraughts."*

We agree with referee #2 in that there are another number of factors that also increase the level of uncertainties. Yet, to improve the representation of the aerosol processes in GCMs is of great importance to adequately simulate aerosol-cloud interactions and their impact in the climatic system. In this case, the suggestions for the modeling of biomass burning

aerosols that arrive from our work are, for the most part, easy to implement, without requiring improvements in the existing parameterizations.

*"Figure 1 is unnecessary to the paper, providing a bit of background context and motivation that can be found elsewhere. At most it is supplementary material or appropriate for an appendix. If it were to remain, I would expect a model sensitivity study to look at the sensitivity of precipitation to mixing state. This would need a much more sophisticated model than used in the current paper.*

*Sections 2.1 to 2.3 do not present any new approaches and can be replaced by a much shorter section, relegating the rest to the Appendix or to supplementary material or simply referenced."*

The manuscript to be submitted for reconsideration was modified accordingly. Figure 1 was removed.

Specific points:

*"i) there can be a strong sensitivity of predicted droplet number to the initial conditions, in particular the height at which an aerosol population is assumed to be in equilibrium with the ambient RH. Table 5 states that the parcel is initiated at 98% RH. Presumably the aerosol populations are assumed to be at equilibrium here. This RH is very close to cloudbase. A mixture of different hygroscopicity of particles will have very different masses of associated water and may have competed for available water more or less successfully already by this stage and may not be at their equilibrium size, dependent on the number of particles in the population. The dependence on initialisation conditions (80, 85, 90, 95, 98, 99% RH, for example) for different updraughts and size distributions may be particularly important for externally-mixed populations. The authors need to demonstrate that 98% is a justifiable initialisation for the entire range of updraughts and particle distributions in their study."*

We thank referee #2 for raising this concern, and will discussed briefly this choice in the article resubmitted. We found that the influence of the initial relative humidity was very low both to supersaturation and to activated fractions. A related Figure was included in the supplementary material.

*"ii) the surface tension of water dependence on temperature may be of some modest importance as Christensen and Petters claim. However, the current manuscript completely ignores the very extensive literature on the roles of surface tension and bulk-to-surface partitioning that has been backwards and forwards in the literature since 1999. This is particularly relevant for particles heavily dominated by the organic components present during biomass burning. The authors need to justify ignoring any discussion or treatment of this, particularly given the recent claims of the pendulum swinging back towards an extremely strong enhancement of activation of organic-rich particles."*

This is an interesting point, and there is, as referee#2 points out, extensive literature on the topic including laboratory data specific for biomass burning that suggest this could be indeed an important issue. However, it was not within the proposed scope of the submitted manuscript to approach this question, considering both the complexity of the biomass burning particles aerosol particles in terms of organic composition. We acknowledged this limitation of the study in the Conclusions.

*"p7 line 11, it is incorrect to state that "McFiggans et al. (2006) proposed sensitivities of the drop number concentration (CCN)..." and then state equation 7. They did propose the method to state sensitivities, but did so with cloud droplet number*

*(N_d). Clearly CCN are not droplets. This sentence can simply be rephrased, but the implications of the underlying understanding of the problem are worrying."*

We thank referee #2 for noting this. We considered activated CCN and cloud droplet number concentration to be similar terms in this initial stages of cloud development. However, the notation we used was confusing and inaccurate at times, when the term "activated" was not included. A better notation was used throughout the text.

[revised manuscript text omitted]

---

## Referee Report (RR1)

**Review of "Impact of mixing state and hygroscopicity on CCN activity of biomass burning aerosol in Amazonia"**

**Sanchez Gacita et al.**

The paper presents results from a cloud parcel model used to simulate the activation of aerosol distributions representative of aerosol from biomass burning in Amazonia. The study presents the sensitivity of the activation of three aerosol populations (representative of moderate to high levels of biomass burning) to uncertainties in assumptions of internal / external mixing, hygroscopicity, kinetic limitations and updraft velocity.

Overall I feel that the paper does not make clear why such a study is required. Much of the sensitivities discussed are already known and have been discussed previously in the literature. Also the link to between this work and previous measurement campaigns that have taken place in this region is a little ambiguous. The wider implications of the work for medium to large scale modelling studies is also not clear.

The paper does present some interesting findings; for example the importance of considering kinetic limitations for particles with moderate / high hygroscopicity. That the treatment of an externally mixed distribution as internally mixed leads to biases in the number of aerosol activated is not surprising, but is clearly demonstrated and may be of interest. Overall the paper is fairly clearly written and plots are clear and useful. The paper would benefit from a proof reading from a native English speaker to correct some minor errors.

Main Comments:

1) I find the description of the aerosol distributions considered quite confusing, in particular the explanation of the difference between Ext1 and Ext2 is hard to follow, this is key and should be made clearer.

    As I understand in Ext1 and Ext2 the authors consider two distributions, one more and one less hydrophilic. Both these distributions comprise an Aitken and an accumulation mode, with fixed size and width. Within a distribution the composition (and therefore hygroscopicity is constant). The authors then alter the overall hygroscopicity of the entire aerosol population by considering scenarios in which the total aerosol number (per mode, per case study) is constant but the relative population of the two distributions is altered.

    This is my understanding, but I think the explanation needs to be made clearer.

2) The ramifications of this work for large scale models are unclear. How do the Ext1 and Ext2 scenarios relate to aerosol schemes used in models? The externally mixed setup used in the paper with (i) a more-hydrophilic and (ii) a less-hydrophilic distribution is the same as that used by many global models (e.g. M7, GLOMAP, EMAC). The scientific impact of the paper can be increased by discussing the findings in relation to the treatment of aerosol in large scale models. Is the treatment in existing models sufficient? Similarly, many of the activation schemes suitable for global models treat kinetic limitations, so are already considering the effects found to be important. If this is the case, then why are the findings of this work important for the modelling community?

3) Is it really the case that biomass burning aerosol from the Amazon has a lower Kappa than other wildfire burning? It seems almost identical to the values from Thailand (Hsiao et al, 2016).

4) No consideration is made of coarse mode aerosol particles. Although small in number, the presence of a few large coarse mode particles could potentially affect the kinetics of the droplet activation, with consequence for the number of activated drops (e.g. Nenes et al, 2001). Would the presence of coarse mode aerosol affect the sensitivities presented? Would consideration of these particles increase the sensitivity to relative humidity?

5) Considering the relative simplicity of the model it's surprising the authors didn't consider more of the parameter space. The authors do not consider sensitivity to particle diameter (which is prognosed in most large-scale models) or mode width (which is not). Would consideration of these affect the conclusions? Why was the choice made to limit these effects?

6) The assumption that composition is independent of size seems a limitation, I understand that this is an idealised study, but from the supplement the Kappa of the accumulation mode is around 20 to 30% larger than that of the Aitken mode. Considering that the sensitivity of $N_d$ to Kappa is largest at this very low hygroscopicity regime it is possible that this effect could be important.

Minor Comments:

1. Table 3. Typo in the temperature: 93K.

2. Pg 1, Line 12. Sentence starting "When the hygroscopicity" is confusing, especially phrase "was supposed to be instead"

3. Pg 5, Line 20: Is there a reference for the cloud parcel model?

4. Pg 5, Line 5: References for models assuming equilibrium.

5. Pg 6, Line 11: Moderately not moderated

6. Pg 9, Line3: Condense not condensate

7. Pg 11, Final sentence: This is confusing! I think you need to be clear in term of recommendations for models whether you are considering freshly emitted aerosol, or mixing with pre-existing aerosol. There seems to be no evidence for your conclusion, and the discussion is suddenly extended from freshly emitted aerosol to mixing with continental aerosol from other sources.

8. Pg 15. The word "situations" isn't quite right. Conditions maybe?

9. Pg 15, line 28: hygroscopicity particles.

10.  Pg 2, Line 31: Pringle et al didn't assume an average hygroscopicity parameter over a single geographical region. The model used is typical of other global models and has two distributions (hydrophillic and hydrophobic) and an individual value of kappa is prognosed for each of the 7 aerosol modes.

Nenes, Athanasios, et al. "Kinetic limitations on cloud droplet formation and impact on cloud albedo." *Tellus B* 53.2 (2001): 133-149. Nenes, Athanasios, et al. "Kinetic limitations on cloud droplet formation and impact on cloud albedo." *Tellus B* 53.2 (2001): 133-149.

---

## Author Response (AR2)

**Response to reviewer #1**

We thank the reviewer #1 for his/her appreciation of the manuscript and for taking the time to make a thorough review and to provide helpful comments that contributed to improve it.

Specific comments and questions raised are addressed below.

*Main Comments:*

*1) Explanation of the difference between Ext1 and Ext2:* Pg 7 Lines 10-23 were modified to refine the explanation of the different external mixing cases considered.

*2) Ramifications of this work for large scale models:*

Our work focused in the study of the impact of hygroscopicity and mixing state for biomass burning aerosols. Some results obtained are likely to have ramifications in global models, like an awareness of the possible bias introduced by the assumption of a single hygroscopicity for biomass burning aerosols around the world and the importance of kinetic limitations as the hygroscopicity increases, but the extension of these findings to multiple populations and conditions that exceed the parameter space considered in our simulations should be done with caution. As the reviewer noted, many activation schemes available to use in GCMs treat kinetic limitations. Yet, the Abdul Razzak Ghan parameterization (Abdul Razzak and Ghan, 2000) does not treat kinetic limitations, and is still in use in many models, including GISS ModelE2 (Bauer et al., 2010, Ban-weiss et al., 2014), GEOS-Chem (Robinson et al., 2007, Pierce et al., 2015), CESM (He et al., 2015) and NorESM (Makkonen et al., 2014). A version modified that includes entrainment is included in the WRF-Chem model (Berg et al., 2015). In addition, physically based parameterizations could soon be coupled to cloud microphysics schemes (Baklanov et al., 2014; Gettelman et al., 2015), as opposite of the current use of look up tables (Thompson and Eidhammer, 2014). Since the microphysics is frequently called, the comparatively high computational cost of parameterizations that account for kinetic limitations could be a limitation. The manuscript was modified, adding the example of the AbdulRazzak parameterization and to include some of these references (pg. 5, lines 15-20).

*3) Is it really the case that biomass burning aerosol from the Amazon has a lower Kappa than other wildfire burning? It seems almost identical to the values from Thailand (Hsiao et al, 2016).*

This is an interesting question and we thank the reviewer for bringing this issue. To the date, data for hygroscopicity of aerosols from open fire biomass burnings is scarce and geographically sparse. $\kappa_P$ values obtained by Rose et at. (2010) and Lathem et al. (2013) (the latter was included in this new version, pg. 4, lines 5-10) are both close to 0.2, which is the defined boundary between low and medium $\kappa_P$ values. The values of $\kappa_P$ compiled in the supplement of this work and those reported by Hsiao et al. (2016) are similar, both in the very low range of $\kappa_P$. Aged aerosols presented hygroscopicities similar or only slightly higher (up to 0.03) than values obtained for recently emitted aerosols (Hsiao et al., 2016; Lathem et al., 2013; Rissler et al., 2004), and particles sampled during the dry season of SMOCC included both recent and aged biomass burning aerosols. Data for $\kappa_P$ of biomass burning aerosols is also frequently obtained from laboratory experiments either in experimental chambers or in small scale open fires (pg 4, lines 1 to 5), reporting a large variability in $\kappa_P$ with values that range from very low to as high as 0.6 for fresh aerosols. Yet, after a short aging this range is reduced and $\kappa_P$ values are in the 0.1-0.3 range (Andreae and Rosenfeld 2008, Engelhart *et al.,* 2012). From this limited set of experimental data, biomass burning aerosols from the Amazon and Thailand appear to have a lower Kappa than other biomass burning aerosols already studied. It

also seems that $\kappa_P$ =0.20 could be considered a reasonable average value. Yet, our results showed that the bias introduced by this assumption when the aerosol population have a very low hygroscopicity could be significant. This is concerning because in some regions where large quantities of biomass burning aerosols are emitted every year the hygroscopicity of the resultant aerosols is unknown or only estimated indirectly. For instance, Vakkari et al. (2014) use a parameterization of $\kappa_P$ in terms of the O/C ratio of the aerosol sample and estimate a range of $\kappa_P$ from 0.11 to 0.21 for samples of biomass burning aerosol collected in Southern Africa.

*4) Consideration of coarse mode aerosol particles.*

The condensation growth of coarse particles is inertially limited, but since their wet size are typically larger than the size of activated particles, so they will are being considered activated as well in this study. Yet, the water condensed on their surfaces might be large enough as to affect the water vapor ratio and cause a drop in the droplet number concentration.
Typically, the ratio of coarse particles number concentration to the total number concentration for biomass burning aerosols is $10^{-4}$ or lower while the mass ratio higher than 0.1 (Janhall et al., 2010). We tested the sensitivity of the relative humidity including a coarse mode with $d_g$ =1.5 µm, $\sigma$ =1.5 and number concentration of 0.6, 6 and 60 for the Case MP$_{1,5}$, resulting in $N_{coarse}/N_{total}$ ratios of $10^{-4}$, $10^{-3}$ and $10^{-2}$.
The result of this test can be seen in Figure S2 of the Supplement. The impact of this coarse mode was very small, most likely due to the low number concentrations. This result agrees well with results by Nenes et al. (2001) for populations with 3 log-normal size distributions, where higher concentrations of coarse particles were considered.

*5) Parameter space:*

The decision of to limit the space parameter of the simulations was made based on two main factors. Firstly, the aerosol size distribution for biomass burning particles is remarkably consistent (Reid et al., 2005). Several observational biomass burning studies conducted in the Amazon region reported rather similar number size distributions for biomass burning aerosols within the boundary layer (Andreae et al., 2004; Artaxo et al., 2013; Brito et al., 2014; Reid et al., 1998; Rissler et al., 2004, 2006). On the other hand, the sensitivity of CCN activation to the aerosol size distribution geometric mean diameter and mode width have been previously estimated and are relatively well established (e.g. McFiggans et al., 2006; Reutter et al., 2009; Ward et al., 2010), so the aerosol size distributions in our study were selected trying to minimize the already known impact of these parameters, and focusing on determining the impact of hygroscopicity and mixing state.

*6) Composition independent of size:*

The effect of this variability of the hygroscopicity with size is indeed expected to influence results, more likely reducing even more the activation in the Aitken mode while increasing the activation in the accumulation mode, while in general decreasing the maximum supersaturations. Yet, even when we acknowledge that the inclusion of this additional level of sophistication would change our numeric results, our conclusions would likely be similar, at least if the consideration of the hygroscopicity change with size is made in a similar way for both hygroscopic groups.

*Minor Comments:*

*1. Table 3. Typo in the temperature:* Corrected in the manuscript to 293 K.

*2. Pg 1, Line 12.* The manuscript was modified to clarify the text.

*3. Pg 5, Line 20: Reference for the cloud parcel model:* This is the first manuscript submitted that make use of the implemented model, so there is no previous reference of it. The approach used, based

on the model described by Prupatcher and Klett (1997), is well known and have been widely used with small modifications that are, for the most part, a consequence of different scientific objectives between studies. Typically, differences between the model implemented for our study and other cloud parcel models based on a similar approach will be found within the following list: treatment of the aerosol population size distribution and the evolution of the population with time, inclusion of external mixing, use of $\kappa_P$ to describe the hygroscopic behavior of aerosols, and approach for the estimation of the activated fraction of the aerosol population.

*4. Pg 5, Line 5: References for models assuming equilibrium.*

The manuscript was modified accordingly (pg. 5 lines 9-12).

*5. Pg 6, Line 11: Moderately not moderated*

Corrected.

*6. Pg 9, Line3: Condense not condensate*

Corrected.

*7. Pg 11, Final sentence.*

We thank the reviewer#1 for noting this inconsistence. Since the aerosol size distribution parameters considered are typical of biomass burning aerosols, our conclusion cannot be extended to aerosols with different size distributions, although a bias in such internal mixture is to be expected. Also, it is unlikely that a global model will make this consideration. The manuscript was modified, removing the last sentence of the paragraph.

*8. Pg 15. The word "situations" isn't quite right. Conditions maybe?*

Corrected.

*9. Pg 15, line 28: hygroscopicity particles.*

Corrected.

*10. Pg 2, Line 31:*

We thank the reviewer for noting that this sentence was confusing. Pringle *et al.* used the EMAC model to simulate the concentration and properties of aerosol particles distributed in 4 hydrophilic and 3 hydrophobic modes externally mixed, and not a single $\kappa_P$ value as could be interpreted from the sentence in the manuscript. Afterwards, bulk aerosol $\kappa_P$ values are estimated assuming internal mixing. Besides comparison with observational data, vertical, horizontal and temporal variability of these bulk aerosol $\kappa_P$ are discussed, and global and regional values are presented.

The sentence was changed to 'Yet, average hygroscopicity parameters have been estimated from both observational and modeled data assuming internal mixing for aerosols from the same emission source (e.g., biomass burning), or even within the same geographical region (Gunthe et al., 2009; Pringle et al., 2010).'

**Response to reviewer #2**

We thank the reviewer #2 for his/her appreciation of the manuscript and for taking the time to review it one more time. Specific comments are addressed below.

*Relative humidity:* We conducted an additional sensitivity test to the initial relative humidity in the simulations adding to the MP1,5 case an additional coarse mode with $d_g$ =1.5 µm, $\sigma$ =1.5 and number concentrations of 0.6, 6 and 60, resulting in $N_{coarse}/N_{total}$ ratios of $10^{-4}$, $10^{-3}$ and $10^{-2}$. The effect of this coarse mode in Again, results exhibited low sensitivity to the relative humidity. We believe this low impact of the relative humidity might be related to the initial temperature of the simulations, since the water vapor mixing ratio is high at 293 K even for an 80% relative humidity.

*Recommendations for GCMs and last paragraph of the conclusion:*

Although GCMs nowadays include activation schemes that account for kinetic limitations, entrainment, giant CCN and other issues, more simpler approach are still in use due to their lower computational cost and comparatively good results in many conditions. The Abdul Razzak Ghan parameterization, for instance, is still in use in models some models, including GISS ModelE2, CESM and NorESM. We agree in that there are other sources of uncertainty that are likely more important, like the sub-grid updraught variability, but in many cases, they are also hard to address, whereas to account for the variability of the hygroscopicity parameter can be accomplished, for instance, modifying the composition of the aerosol population regionally in the emissions.

*Surface properties*: We modified the conclusions addressing this limitation (pg. 16, lines 1-4).

*Abstract*: The abstract was modified to clarify conclusions and improve legibility.

**Additional bibliography**

Artaxo, P., Rizzo, L. V., Brito, J. F., Barbosa, H. M. J., Arana, A., Sena, E. T., Cirino, G. G., Bastos, W., Martin, S. T. and Andreae, M. O.: Atmospheric aerosols in Amazonia and land use change: from natural biogenic to biomass burning conditions, Faraday Discuss., doi:10.1039/c3fd00052d, 2013.

Berg, L. K., Shrivastava, M., Easter, R. C., Fast, J. D., Chapman, E. G., Liu, Y. and Ferrare, R. A.: A new WRF-Chem treatment for studying regional-scale impacts of cloud processes on aerosol and trace gases in parameterized cumuli, Geosci. Model Dev., 8(2), 409–429, doi:10.5194/gmd-8-409-2015, 2015.

Gettelman, A., Morrison, H., Santos, S., Bogenschutz, P. and Caldwell, P. M.: Advanced two-moment bulk microphysics for global models. Part II: Global model solutions and aerosol-cloud interactions, J. Clim., 28(3), 1288–1307, doi:10.1175/JCLI-D-14-00103.1, 2015.He, J., Zhang, Y., Glotfelty, T., He, R., Bennartz, R., Rausch, J. and Sartelet, K.: Decadal simulation and comprehensive evaluation of CESM/CAM5.1 with advanced chemistry, aerosol microphysics, and aerosol-cloud interactions, J. Adv. Model. Earth Syst., 7, 110–141, doi:10.1002/2014MS000360, 2015.

Janhall, S., Andreae, M. O. and Posch, U.: Biomass burning aerosol emissions from vegetation fires : particle number and mass emission factors and size distributions, Atmos. Chem. Phys, 10, 1427–1439, 2010.

Reid, J. S., Hobbs, P. V., Ferek, R. J., Blake, D. R., Martins, J. V., Dunlap, M. R. and Liousse, C.: Physical, chemical, and optical properties of regional hazes dominated by smoke in Brazil, J. Geophys. Res, 103(98), 1998.

Reid, J. S., Koppmann, R., Eck, T. F. and Eleuterio, D. P.: A review of biomass burning emissions part II: intensive physical properties of biomass burning particles, Atmos. Chem. Phys, 5, 799–825, doi:10.5194/acp-5-

799-2005, 2005.

Thompson, G. and Eidhammer, T.: A study of aerosol impacts on clouds and precipitation development in a large winter cyclone, J. Atmos. Sci., (2012), 140507124141006, doi:10.1175/JAS-D-13-0305.1, 2014.

Vakkari, V., Kerminen, V.-M., Beukes, J. P., Titta, P., Zyl, P. G. van, Josipovic, M., Wnter, A. D., Jaars, K., Worsnop, D. R., Kulmala, M. and Laakso, L.: Geophysical Research Letters, Geophys. Res. Lett., 2644–2651, doi:10.1002/2014GL059396.Received, 2014.

[revised manuscript text omitted]